# Overfitting Behaviour of Gaussian Kernel Ridgeless Regression: Varying Bandwidth or Dimensionality

**Marko Medvedev**
The University of Chicago
medvedev@uchicago.edu

**Gal Vardi**
Weizmann Institute of Science
gal.vardi@weizmann.ac.il

**Nathan Srebro**
TTI-Chicago
nati@ttic.edu

## Abstract

We consider the overfitting behavior of minimum norm interpolating solutions of Gaussian kernel ridge regression (i.e. kernel ridgeless regression), when the bandwidth or input dimension varies with the sample size. For fixed dimensions, we show that even with varying or tuned bandwidth, the ridgeless solution is never consistent and, at least with large enough noise, always worse than the null predictor. For increasing dimension, we give a generic characterization of the overfitting behavior for any scaling of the dimension with sample size. We use this to provide the first example of benign overfitting using the Gaussian kernel with sub-polynomial scaling dimension. All our results are under the Gaussian universality ansatz and the (non-rigorous) risk predictions in terms of the kernel eigenstructure.

## 1 Introduction

A central question in learning theory is how learning algorithms can generalize well even when returning models that perfectly fit (i.e. interpolate) noisy training data. This phenomenon was observed empirically by Zhang et al. [63], and does not align with the traditional belief from statistical learning theory that overfitting to noise leads to poor generalization. Consequently, it attracted significant interest in recent years, and there has been much effort to understand the overfitting behavior of linear models, kernel methods, and neural networks.

In this paper, we study the overfitting behavior of Kernel Ridge Regression (KRR) with Gaussian kernel, namely, the behavior of the limiting test error when training on noisy data as the number of samples tends to infinity by insisting on interpolation (achieving zero training error). When the input dimension and bandwidth are fixed, the overfitting behavior is known to be "catastrophic" [37], i.e. for any nonzero noise, the test risk tends to infinity as the sample size increases. However, this is not how Gaussian Kernel Ridge Regression is typically used in practice. In fixed dimension, the bandwidth is tuned, that is decreased, when the sample size increases [51, 19]. Additionally, it makes sense to study the behaviour when the input dimension increases with sample size (as in, e.g. linear models with proportional scaling [42]). This could be because when more data is available, more input features are used, even with a kernel; because as more resources are available we scale up both the input dimension and amount of data used; or to capture the fact that very large scale problems typically involve both more samples and higher input dimension. But unlike with linear models, where the dimension must scale linearly with the number of samples in order to allow for interpolation, when a kernel is used we can study the behaviour even when the input dimensionality increases much slower, and ask how slowly it could increase without catastrophic overfitting. Previous studies on kernel ridgeless regression considered polynomial increasing dimension (i.e. dimension $\propto$ sample-size$^a$, for $0 < a \leq 1$) [4, 64, 24, 38, 40], but not subpolynomial scaling.

We aim to provide a more comprehensive picture of overfitting with Gaussian KRR by studying the overfitting behavior with varying bandwidth or with arbitrarily varying dimension, including

38th Conference on Neural Information Processing Systems (NeurIPS 2024).

sub-polynomially. In particular, we show that for fixed dimension, even with varying bandwidth, the interpolation learning is never consistent and generally not better than the null predictor (either the test error tends to infinity or is finite but it is almost always not better than the null predictor). For increasing dimension, we give an upper and lower bound on the test risk for any scaling of the dimension with sample size, which indicates in many cases whether the overfitting is catastrophic (test error tends to infinity), tempered (test error tends to a constant), or benign (consistent). Our result agrees with the polynomial scaling of the dimension with sample size, showing tempered overfitting for an exponent that is a reciprocal of an integer and benign overfitting for any other exponent [4, 64]. Moreover, our result goes further, as we show the first example of sub-polynomially scaling dimension that achieves benign overfitting for the Gaussian kernel. Additionally, we show that a class of dot-product kernels on the sphere is inconsistent when the dimension scales logarithmically with sample size. All our results are under the Gaussian universality ansatz and the non-rigorous but well-established risk predictions in terms of the kernel eigenstructure [50, 66, 8, 57].

### Related work

The test performance of overfitting models has been extensively studied for linear regression [27, 6, 3, 43, 45, 16, 32, 58, 53, 65, 30, 54, 13, 2, 49, 25], linear classification [12, 55, 10, 44, 42, 49, 36, 52, 56, 18], neural networks [20, 21, 11, 34, 60, 61, 39, 33, 23, 31, 29], and kernel methods. Below we focus on overfitting in kernel ridge regression.

**Overfitting in fixed dimension with fixed kernel.** In Mallinar et al. [37], the authors show that the minimum norm interpolating solution for Gaussian kernel with fixed bandwidth overfits catastrophically. In [15, 4, 64], the authors derive bounds on the test risk of minimum norm interpolant with a fixed kernel under various assumptions. Our result for fixed dimension will only apply to the Gaussian kernel, but it allows for any varying or adaptively chosen bandwidth.

**Inconsistency of Kernel Ridge Regression.** The case of varying bandwidth has been considered in Beaglehole et al. [5], Rakhlin and Zhai [47], Haas et al. [26]. In Beaglehole et al. [5], the authors show that there exists a specific data distribution for which the minimum norm interpolanting solution for a particular set of translation invariant kernels is not consistent. In Rakhlin and Zhai [47], the authors show that for input distributions on the unit ball, the Laplace kernel is inconsistent, even with varying bandwidth. In Haas et al. [26], the authors show that under different assumptions on the data distribution, for a general class of (potentially varying) kernels in fixed dimension, any differentiable function that overfits the data and is not much different from the minimum norm interpolant is inconsistent. All of these works only consider whether we can achieve consistency. None of these results apply in the case of data distributions that we are considering, and even if the predictor is not consistent, we ask how bad is it by comparing it to the null predictor and whether it might be tempered.

**Overfitting in increasing dimensions.** Many papers have studied the setup where the dimension increases with sample size [4, 64, 66, 38, 40, 28, 59], in particular when the dimension is a function of sample size (or vice versa), but they all consider only the case of a polynomial scaling of dimension and sample size. It was shown that in this case the minimum norm interpolating solution of dot product kernels on the sphere can be benign, depending on if the exponent is not an integer. We generalize these results to any scaling of the dimension and sample size. Our results recover the existing results in the case of polynomially scaling dimension and show benign overfitting in a certain sub-polynomial scaling. Having sub-polynomimal scaling of the dimension allows us to expand the set of possible target functions from only polynomials of a bounded degree, as in the case of polynomial scaling of dimension [4, 64, 66, 38, 40], to, in our case of sub-polynomially increasing dimension, polynomials of any degree and even non-polynomial functions.

## 2 Problem formulation and assumptions

**Kernel ridge(less) regression and the Gaussian kernel.** Let $\mathcal{D}$ be an unknown distribution over $\mathcal{X} \times \mathcal{Y} \subseteq \mathbb{R}^d \times \mathbb{R}$ and let $\{(x_i, y_i)\}_{i=1}^m \sim \mathcal{D}^m$ be a dataset consisting of $m$ samples. For simplicity, we will assume that the distribution of the target is given by a target function $f^*$ of the input $x \in \mathcal{X}$

with zero mean independent noise $\xi$ with variance $\sigma^2$, that is $y \sim f^*(x) + \xi$. We note that our results can be extended to a distribution agnostic setting, as analyzed by Zhou et al. [66].

Let $K : \mathcal{X} \times \mathcal{X} \to \mathbb{R}$ be a positive semi-definite kernel function. Let $\|f\|_K$ be the norm of $f$ in the RKHS $\mathcal{H}_K$ corresponding to $K$. For a predictor $f$, let $R(f)$ and $\hat{R}(f)$ be the test and training risk of $f$,

$$\hat{R}(f) = \frac{1}{m} \sum_{i=1}^{m} (f(x_i) - y_i)^2 \ \text{ and } \ R(f) = \mathbb{E}_{\mathcal{D}} \left[ (f(x) - y)^2 \right].$$

Two important risks to consider are the risk of the null predictor, $f \equiv 0$, which we will denote by $R(0) = \mathbb{E}_{\mathcal{D}}[(y)^2]$, and the Bayes (or irreducible) risk, which we will denote $\sigma^2$ or $R(f^*)$. Using this notation, the risk of the null predictor is $R(0) = \sigma^2 + \mathbb{E}_{\mathcal{D}}[(y)^2]$. Bayes risk represents the minimum possible risk that can be achieved by any predictor. For a regularization parameter $\delta$, the regularized ridge solution $\hat{f}_\delta$ is given by

$$\hat{f}_\delta = \text{argmin}_{f \in \mathcal{H}_K} \hat{R}(f) + \frac{\delta}{m} \|f\|_K^2.$$

We are interested in the minimum norm interpolating (ridgeless) solution $\hat{f}_0 = \lim_{\delta \to 0+} \hat{f}_\delta$, namely

$$\hat{f}_0 = \text{argmin}_{\hat{R}(f)=0; f \in \mathcal{H}_K} \|f\|_K^2.$$

We will focus on the Gaussian kernel, which is given by $K(x_1, x_2) = \exp\left(-\frac{\|x_1 - x_2\|_2^2}{\tau_m^2}\right)$, where $m$ is the sample size and $\tau_m$ is a predetermined bandwidth parameter that can vary with sample size. The Gaussian kernel is widely used and achieves good error rates for a variety of learning tasks [51, 19]. Gaussian KRR achieves optimal convergence to the best possible (Bayes) error for learning any function in a Besov space of high enough order (essentially bounded and twice differentiable in the weak sense) under very mild assumptions on the distribution of the input and target $\mathcal{X} \times \mathcal{Y}$ [19]. For ridge regression with the Gaussian kernel and under a standard data distribution assumption, the minimum distance between samples decreases with sample size so it makes sense to also decrease $\tau_m$. Additionally, decreasing $\tau_m$ with sample size helps to achieve good convergence rates theoretically [19].

**Main question.** We will consider the problem of learning using the minimum norm interpolating solution $\hat{f}_0$ of KRR. We want to understand the limiting behavior of test risk $R(\hat{f}_0)$ as the sample size increases $m \to \infty$, that is $\lim_{m \to \infty} R(\hat{f}_0)$. It suffices to understand $\lim\sup_{m \to \infty} R(\hat{f}_0)$ and $\lim\inf_{m \to \infty} R(\hat{f}_0)$ and this way we do not assume the existence of the limit. In this work, we use the taxonomy of benign, tempered, and catastrophic overfitting from Mallinar et al. [37], which indicates whether $\lim_{m \to \infty} R(\hat{f}_0)$ is the Bayes (optimal) error, a non-optimal but constant error, or infinity. Note that in this taxonomy, the null predictor can be classified as tempered. Therefore, we will compare the limiting risk to the risk of the null predictor $R(0)$ in order to understand whether the performance of the interpolating solution is non-trivial.

**Main tool: Eigenframework and a closed form of the test risk.** Our main tool will be the closed form of the test risk predicted by the *eigenframwork* [50]. Under the eigenframework, we can write down the closed form of the test risk using Mercer's theorem decomposition of a kernel function $K$.

Given a positive semi-definite kernel function $K : \mathcal{X} \times \mathcal{X} \to \mathbb{R}$, we can decompose it as

$$K(x_1, x_2) = \sum_{k=1}^{\infty} \lambda_k \phi_k(x_1) \phi_k(x_2), \tag{1}$$

where $\lambda_k$ and $\phi_k$ are the eigenvalues and eigenfunctions of the integral operator associated to $K$. The eigenfunctions $\{\phi_k\}$ are an orthonormal basis of $L^2_{\mathcal{D}_\mathcal{X}}(\mathcal{X})$, where $\mathcal{D}_\mathcal{X}$ is the marginal distribution of $\mathcal{X}$. We denote the Bayes optimal target function by $f^*$, and expand it in the kernel basis $\{\phi_i\}_{i=1}^{\infty}$ as $f^*(x) = \sum_{i=1}^{\infty} \beta_i \phi_i(x)$. To state the close form of the test risk we will introduce a few quantities. Let *effective regularization*, $\kappa_\delta$, be the solution to $\sum_{i=1}^{\infty} \frac{\lambda_i}{\lambda_i + \kappa_\delta} + \frac{\delta}{\kappa_\delta} = m$. Furthermore, let $\mathcal{L}_{i,\delta} = \frac{\lambda_i}{\lambda_i + \kappa_\delta}$

and $\mathcal{E}_\delta = \frac{m}{m - \sum_{i=1}^\infty \mathcal{L}_{i,\delta}^2}$. Then, the *predicted risk*, i.e. the predicted closed form of the test risk of $\hat{f}_\delta$, is given by

$$\tilde{R}(\hat{f}_\delta) = \mathcal{E}_\delta \left( \sum_{i=1}^\infty \left(1 - \mathcal{L}_{i,\delta}\right)^2 \beta_i^2 + \sigma^2 \right), \tag{2}$$

where $\sigma^2$ is the Bayes error of $\mathcal{D}$. Equation (2) was initially heuristically derived using the replica method or continuous approximations to the learning curves inspired by the Gaussian process literature [7]. In [50], it is dervied using a conservation law. Note that [66] shows that the predicted closed form of the test risk from [50] extends to general target distributions. There is strong evidence that the predicted risk is a good estimate of the true test risk, namely $R(\hat{f}_\delta) \approx \tilde{R}(\hat{f}_\delta)$. Indeed, a number of works use the predicted risk closed form to estimate the test risk of KRR [58, 8, 66]. The following assumption on Gaussian design ansatz is used by all of these works.

**Assumption 1** (Gaussian design ansatz, cf. Zhou et al. [66]). When sampling $(x, \cdot) \sim \mathcal{D}$, we have that the Gaussian universality holds for the eigenfunctions in the sense that the expected risk is unchanged if we replace $\phi$ with $\tilde{\phi}$, where $\tilde{\phi}$ is Gaussian with appropriate parameters, i.e. $\tilde{\phi} \sim \mathcal{N}(0, \text{diag}\{\lambda_i\})$.

This assumption appears to hold for real datasets as well, namely, the predictions computed for Gaussian design agree well with the experiments on kernel regression using real data [8, 50, 57]. As discussed in Zhou et al. [66], under this assumption, the equivalence $R(\hat{f}_\delta) \approx \tilde{R}(\hat{f}_\delta)$ holds in a few ways. First, in an appropriate asymptotic limit in which the sample size $m$ and the number of eigenmodes in a given eigenvalue grow proportionally, the equivalence holds [27, 1]. Second, if the eigenstructure of the task is fixed, the error between the two can be bounded by a decaying function of $m$ [14]. Finally, various numerical experiments show that the error between the two is small even for a small sample size $m$ [8, 50]. Specifically, Canatar et al. [8] (see Figure 5 there) gives empirical evidence that the predicted risk closely approximates the true risk for the Gaussian kernel with data uniform on a sphere, which is the setting that we consider in some of our results.

There has been some recent progress in bounding the error between $R(\hat{f}_\delta)$ and $\tilde{R}(\hat{f}_\delta)$ unconditionally. Misiakiewicz and Saeed [41] shows that the error will tend to zero if the dimension $d$ grows fast enough with the sample size. Additionally, they provide strong empirical evidence that the predicted risk is close to the test risk for a real dataset (MNIST) and Gaussian kernel, see Figure 1 in [41].

Formally, we will prove results about the predicted risk $\tilde{R}(\hat{f}_\delta)$, but as previously presented evidence suggests, treating $R(\hat{f}_\delta) \approx \tilde{R}(\hat{f}_\delta)$ as equivalence is sufficient for understanding the behavior of KRR.

We note that using the eigenframework might introduce restrictions for which kernels some of our results apply, as concurrent work showed that the eigenframework prediction might not hold for the NTK in fixed dimension [4, 15]. Our two main results concern the Gaussian kernel, for which we described ample empirical evidence that the eigenlearning predictions hold [8, 41]. Understanding the limitations of the eigenframework is an important future research direction.

## 3 Fixed dimension: Gaussian kernel with varying bandwidth

We will assume that the source distribution is uniform on a $d$ dimensional sphere, that the target function is square integrable, and that the target distribution is given by the target function with an independent noise.

**Assumption 2** (Target function and data distribution). Let $\mathcal{D}$ be the distribution over $\mathcal{X} \times \mathcal{Y} = \mathbb{S}^{d-1} \times \mathbb{R}$, such that the $\mathcal{X}$ marginal, denoted by $\mathcal{D}_{\mathcal{X}}$, is $\text{Unif}(\mathbb{S}^{d-1})$. We will assume that for a target function $f^* \in L^2_{\mathcal{D}_{\mathcal{X}}}(\mathbb{S}^{d-1})$, the marginal $\mathcal{Y}$ distribution is given by $y \sim f^*(x) + \xi$, where $\xi$ has mean zero and variance $\sigma^2 > 0$. We write $f^* = \sum_i \beta_i \phi_i$, where $\{\phi_i\}$ is the $L^2_{\mathcal{D}_{\mathcal{X}}}(\mathbb{S}^{d-1})$ eigenbasis corresponding to the kernel $K$ (Equation (1)). If we write $\beta = (\beta_1, \beta_2, \dots)$, then we have that $\|\beta\|_2^2 = \mathbb{E}_{\mathcal{D}_{\mathcal{X}}}\left((f^*(x))^2\right)$. We will use the notation $\|f^*\|^2 = \|\beta\|_2^2$.

The assumption on the distribution on $\mathcal{X}$ is common in the literature on KRR [38, 4, 64, 22]. The assumption that $x \sim \text{Unif}\left(\mathbb{S}^{d-1}\right)$ can be relaxed to a more general setting where $x$ is uniformly distributed on other manifolds that are diffeomorphic to the sphere using the results of Li et al. [35], although that will not be the focus of this paper.

Note that here we vary the bandwidth $\tau_m$ so both $\hat{f}_0$ and $R(\hat{f}_0)$ will depend on the bandwidth $\tau_m$ as well as $m$. We will identify three different regimes of bandwidth scaling. We will show that the minimum norm interpolating solution exhibits either tempered or catastrophic overfitting, and we will argue that it is almost always worse than the risk of the null predictor.

**Theorem 3** (Overfitting behavior of Gaussian kernel in fixed dimension). Under Assumption 2, the following bounds hold for the predicted risk $\tilde{R}(\hat{f}_0)$ of the minimum norm interpolating solution of Gaussian KRR:

1. If $\tau_m = o(m^{-\frac{1}{d-1}})$, then $\tilde{R}(0) \leq \liminf_{m \to \infty} \tilde{R}(\hat{f}_0) \leq \limsup_{m \to \infty} \tilde{R}(\hat{f}_0) < \infty$. More precisely, if $\tau_m \leq m^{-\frac{1}{d-1}} t(m)$, where $t(m) \to 0$ as $m \to \infty$, then there is a scalar $c_d$ that depends only on the dimension and $m_0$ that depends on $t(m)$ such that for all $m > m_0$ we have $\tilde{R}(\hat{f}_0) > \sigma^2 + (1 - c_d t(m)^{\frac{d-1}{2}}) \|f^*\|^2$.

2. If $\tau_m = \omega(m^{-\frac{1}{d-1}})$, then $\lim_{m \to \infty} \tilde{R}(\hat{f}_0) = \infty$. Hence, for large enough $m$ we have $\tilde{R}(\hat{f}_0) > \tilde{R}(0)$.

3. If $\tau_m = \Theta(m^{-\frac{1}{d-1}})$, then $\limsup_{m \to \infty} R(\hat{f}_0) < \infty$. Moreover, Suppose that $C_1 m^{-\frac{1}{d-1}} \leq \tau_m \leq C_2 m^{-\frac{1}{d-1}}$ for some constants $C_1$ and $C_2$, then there exist $\eta, \mu > 0$ that depend only on $d, C_1$, and $C_2$, such that for all $m$ we have $\tilde{R}(\hat{f}_0) > \mu\|f^*\|^2 + (1+\eta)\sigma^2$. Consequently, $\tilde{R}(\hat{f}_0) > \tilde{R}(0)$ as long as $\sigma^2 > \frac{1-\mu}{\eta}\|f^*\|^2$.

Theorem 3 shows that the minimum norm interpolating solution of Gaussian KRR cannot be consistent when data is distributed uniformly on the sphere, even with varying or adaptively chosen bandwidth. Additionally, in the first two modes of bandwidth change, the minimum norm interpolating solution is never better than the null predictor. In the third case, the interpolating solution is worse than null for noise that is not too small. This shows that even though the minimum norm interpolating predictor is classified as tempered in the first and third cases of scaling of the bandwidth, it is still worse than the trivial null predictor. Note that our analysis does not exclude the possibility that for $\tau_m = \Theta(m^{\frac{1}{d-1}})$ there exists small enough $\sigma^2$ for which the interpolating solution is better than the null predictor. We leave this as an open question. In Appendix A, we provide further empirical justification for Theorem 3.

## 4 Increasing dimension

For the case of increasing dimension, we consider the problem of learning a sequence of distributions $\mathcal{D}^{(d)}$ over $\mathcal{X} \times \mathcal{Y} = \mathbb{R}^d \times \mathbb{R}$ given by $y \sim f_d^*(x) + \xi_d$ using a sequence of kernels $K^{(d)}$. Here, $\xi_d$ is independent noise with mean 0 and variance $\sigma^2 > 0$. Formally, the kernel and the target function can change with the dimension $d$, but we will think of it as the same kernel and target with higher dimensional input. Furthermore, $d$ will increase with sample size $m$, i.e. $d = d(m)$ (or analogously $m$ will increase with $d$). A common assumption, which we also adopt, is that the projections of the target function $f_d^*$ onto the eigenfunctions $\phi_k^{(d)}$ of the kernels $K^{(d)}$ are uniformly bounded [4, 64].

**Assumption 4** (Target function and distribution in increasing dimension). Consider learning a sequence of target functions $f_d^*$ with a sequence of kernels $K^{(d)}$. Let the target function $f_d^*$ have only $S_d$ nonzero coefficients (where $S_d$ can change with $d$), so $f_d^* = \sum_{i=1}^{S_d} \beta_i^{(d)} \phi_i^{(d)}$ where $\phi_i^{(d)}$ are from Equation (1) and $|\beta_i^{(d)}| \leq B$, i.e. $\|\beta\|_\infty \leq B$, for $B$ that is independent of $d$.

The functions that can be represented in this form depend on the number of nonzero coefficients, $S_d$, and the kernel that we are using. In particular, for dot-product kernels on the sphere it includes all polynomials of degree $k \leq k_d$ where $k_d$ is such that the multiplicities of first $k_d$ eigenfunctions are at most $S_d$. If $S_d$ grows with $d$, then this set will include much more general functions. See Remark 14 for further discussion.

First, we will consider a general kernel $K$ since Theorem 7 and Theorem 9 will apply more generally. Then, we will apply these results to the cases of dot-product and Gaussian kernels.

The multiplicities of eigenvalues will play an important role in bounding the test risk, both from above and below, so we will introduce the following related notation.

**Definition 5** (Lower and upper index). Let $\tilde{\lambda}_k$ be the $k$-th non-repeating eigenvalue of a kernel $K$ and let $N(k)$ be its multiplicity. Let $N_k = N(1) + \cdots + N(k)$. Let $m$ be the sample size. Let $k_m$ be defined as the maximal $k$ such that there is less than $m$ eigenvalues with index $k$, i.e. $k_m = \max\{k \in \mathbb{N} | N(1) + \cdots + N(k) < m\}$. Define the lower index $L_m$ and the upper index $U_m$ as follows $L_m = N(1) + \cdots + N(k_m)$ and $U_m = N(1) + \cdots + N(k_m + 1)$. When the dimension changes with sample size, we sometimes denote $N(k)$ by $N(d, k)$.

We will first state a generic bound on the test risk for any data distribution, kernel with a bounded sum of eigenvalues, sample size, and dimension. This bound will be informative when we scale the dimension $d$ with sample size $m$, but it holds for any kernel that satisfies the following assumption.

**Assumption 6** (Bounded sum of eigenvalues). Assume that the kernel $K$ has a bounded sum of eigenvalues, i.e. there is a constant $A$ such that $\sum_{i=1}^{\infty} N(i)\tilde{\lambda}_i \leq A$. For a sequence of kernels $K^{(d)}$, assume that all such $A^{(d)}$ are bounded by some constant $A$.[1]

This is a reasonable assumption for most dot-product kernels, as we show in Appendix C.4. It also implicitly sets the scale of the kernel.

**Theorem 7** (Test risk upper bound for kernel ridgeless regression). Let $d$ and $m$ be any dimension and sample size. Define $L_m, U_m, k_m, N(i), N_l$, and $\tilde{\lambda}_k$ as in Definition 5. Consider KRR with a kernel $K$ satisfying Assumption 6 for some $A$. Assume that for some integer $l$, the target function $f^*$ satisfies Assumption 4 with at most $N_l$ nonzero coefficients. Then, the predicted risk of the minimum norm interpolating solution is bounded by the following:

$$\tilde{R}(\hat{f}_0) \leq \left(1 - \frac{L_m}{m}\right)^{-1} \left(1 - \frac{m}{U_m}\right)^{-1} \sigma^2 \tag{3}$$

$$+ B^2 \left(1 - \frac{L_m}{m}\right)^{-1} \left(1 - \frac{m}{U_m}\right)^{-1} \frac{A^2}{m^2} \left(\sum_{i=1}^{l} N(i)\frac{1}{\tilde{\lambda}_i^2}\right). \tag{4}$$

Alternatively, we can bound the risk using $\left(\sum_{i=1}^{l} N(i)\frac{1}{\tilde{\lambda}_i}\right)$ instead of $\left(\sum_{i=1}^{l} N(i)\frac{1}{\tilde{\lambda}_i^2}\right)$ (see Theorem 20 in the appendix).

We also establish a generic inconsistency result for any data distribution, kernel with a bounded sum of eigenvalues, sample size, and dimension based on the upper and lower indices from Definition 5. We will further need to assume that the eigenvalues are bounded away from zero. Similarly, this will be useful when scaling dimension $d$ with sample size, but it holds generally.

**Assumption 8** (Lower bound on eigenvalues). Assumme that the kernel $K$ has eigenvalues that are not too small, i.e. there is a constant $b$ such that $\max_{i \leq k_m} \left(\frac{1}{\tilde{\lambda}_i}\right) < \frac{m - L_m}{b}$. For a sequence of kernels $K^{(d)}$, assume that for the corresponding $m = m(d)$ (since $d = d(m)$, we can also "invert" the dependence) all such $b^{(d)}$ are bounded below by some $b$.

This assumption will hold for most dot-product kernels and we will show it for Gaussian kernel in Appendix C.4.

**Theorem 9** (Test risk lower bound for any kernel ridgeless regression). Let $k_m$ and $L_m$ be as in Definition 5. Consider learning a target function $f^*$, with some sample size $m$. Let $K$ be a kernel satisfying Assumption 6 and Assumption 8 for some $A$ and $b$. Consider the minimum norm interpolating solution of KRR (with any data distribution) with kernel $K$. Then, for the predicted risk of minimum norm interpolating solution, the following lower bound holds:

$$\tilde{R}(\hat{f}_0) > \left(1 - \left(\frac{b}{b+1}\right)^2 \frac{L_m}{m}\right)^{-1} \sigma^2.$$

To apply Theorem 7 for varying dimension $d$, we would additionally require that $A$ is uniformly bounded for all $d$ and kernel $K^{(d)}$ and also that $l = l(d)$ changes with $d$ such that Assumption 4 holds with $S_d = N_{l(d)}$. For dot-product kernels $K^{(d)}$ on the sphere, if we let $K^{(d)}(x, y) = h^{(d)}(\|x - y\|)$,

---

[1]Note that this assumption implicitly sets the scale of the kernel.

we will have $A = \sup_d h^{(d)}(0)$, so if $h^{(d)}$ does not change with $d$ we can take $A = h(0)$ (see Appendix C.4 for more details). Specifically, this holds for the Gaussian kernel on the sphere with $A = 1$.

To apply Theorem 9 to the case of increasing dimension, we would require that the bounds $\sum_i N(d,i)\tilde{\lambda}_i \leq A$ and $\max_{i\leq k_m}\left(\frac{1}{\tilde{\lambda}_i}\right) < \frac{m-L_m}{b}$ hold for all $d$ and kernels $K^{(d)}$. Usually, the condition $\max_{i\leq k_m}\left(\frac{1}{\tilde{\lambda}_i}\right) < \frac{m-L_m}{b}$ will be satisfied for $b = 1$. We will show it for the two cases of sub-polynomially scaling dimensions with a Gaussian kernel. For the polynomial scaling dimension, it is reasonable to assume it for general dot-product kernels, as discussed in Appendix C.4.

Now, we will show that using Theorem 7 and Theorem 9 we can recover the behavior of the minimum norm interpolating solution for polynomially increasing dimension [4, 64], i.e. tempered overfitting for integer exponent and benign for non-integer exponent. Here, we will need to additionally assume that the eigenvalue decay is not too fast.

**Assumption 10** (Eigenvalue decay). The eigenvalues do not decrease too quickly, i.e. for $k_m$ as in Definition 5, we have that there is a constant $c$ such that $\max_{i\leq k_m}\left(\frac{1}{\tilde{\lambda}_i}\right) \leq cN(k_m)$. For increasing dimension, we require that $\max_{i\leq k_m}\left(\frac{1}{\tilde{\lambda}_i}\right) \leq cN(d,k_m)$ for all $m$ (i.e. all $d$, as $d$ and $m$ both increase).

This assumption is stronger than Assumption 8, but as we show in Appendix C.4, it is reasonable for dot-product kernels on the sphere and even the NTK.

**Corollary 11** (Dot-product kernels with polynomially increasing dimension, recovering the results of [24, 38, 40, 4, 64]). Consider the problem of learning a sequence of target functions $f_d^*$ satisfying Assumption 4 with $S_d \leq \Theta(d^{\lfloor\alpha\rfloor})$ with a dot-product kernel $K(x,y) = h(\|x-y\|)$ with $h(0) = 1$ on the sphere $\mathbb{S}^{d-1}$ (where $h$ does not depend on $d$, i.e. $A = 1$ from Assumption 6)) that further satisfies Assumption 10. Let $\frac{d^\alpha}{m} = \Theta(1)$ for $\alpha \in (0,\infty)$. Then the overfitting behavior of the minimum norm interpolating solution is benign if $\alpha$ is not an integer and tempered if $\alpha$ is an integer.

Additionally, we will show that for $d = \log m$, we cannot get benign overfitting, i.e. consistency with a class of dot-product kernels on the sphere. Similarly, as in the previous corollary, this will hold for any sequence where $d = \log m$ even only asymptotically.

**Corollary 12** (Inconsistency with dot-product kernels in logarithmically scaling dimension). Let $K^{(d)}$ be a sequence of dot-product kernels on $\mathbb{S}^{d-1}$ that satisfy Assumption 8. Let the dimension $d$ grows with sample size as $d = \log_2 m$ (i.e. $m = 2^d$). Then, the minimum norm interpolant cannot exhibit benign overfitting for any such sequence $K^{(d)}$, i.e. there exists an absolute constant $\eta > 0$ such that for all $d, m$, $\tilde{R}(\hat{f}_0) > (1 + \eta)\sigma^2$.

On the other hand, using Theorem 7, we will establish the first case of sub-polynomial scaling dimension with benign overfitting using the Gaussian kernel and data on the sphere. We will use $d = \exp(\sqrt{\log m})$.

**Corollary 13** (Benign overfitting with Gaussian kernel and sub-polynomial dimension). Let $K$ be the Gaussian kernel on the sphere $\mathbb{S}^{d-1}$ with a fixed bandwidth, and take a sequence of dimensions $d$ and sample sizes $m$ that scale as $d = \exp\left(\sqrt{\log m}\right)$ (in particular, we take $l \in \mathbb{N}$ such that $d = 2^{2^l}$ and $m = 2^{2^{2l}}$ with $l = 1,2,3\dots$). Consider learning a sequence of target functions $f_d^*$ as in Assumption 4 with $S_d \leq m^{\frac{1}{4}}$. Then, we have that the minimum norm interpolating solution achieves the Bayes error in the limit $(m,d) \to \infty$. In particular, for $d \geq 4$ and $m \geq 16$ we have

$$\tilde{R}(\hat{f}_0) \leq \left(1 - \frac{1}{\log m}\right)^{-1}\left(1 - \exp\left(-0.89\sqrt{\log m}\right)\right)^{-1}\sigma^2 + 2B^2\frac{1}{m}.$$

**Remark 14** (Allowed target functions). The set of allowed target functions $f_d^*$ in Corollary 13, i.e. with sub-polynomial scaling dimension, is strictly larger than the set of allowed target functions for polynomially scaling dimension, as in Corollary 11 and [38, 4, 64]. In particular, for polynomially scaling dimension $\frac{d^\alpha}{m} = \Theta(1)$, the result holds only if the target function is a polynomial of degree at most $\lfloor\alpha\rfloor$. On the other hand, Corollary 13 shows that sub-polynomially scaling dimension allows for the target function to be a polynomial of arbitrary degree, as well as non-polynomial functions. In particular, in dimension $d$, we can represent polynomials of degree up to $\Theta(\log^2 d)$.

# 5 Proofs outline

In this section, we discuss the main proof ideas. All formal proofs are provided in the appendix.

By Equation (2), to understand how the test risk of the minimum norm interpolating solution behaves, it suffices to understand how the eigenvalues corresponding to the kernel $K$ and thus the quantities $\mathcal{E}_0, \mathcal{L}_{i,0}$ behave. In Zhou et al. [66], the authors show that $\mathcal{E}_0$ is bounded both above and below in terms of the effective rank of the systems of eigenvalue $\{\lambda\}_{i=1}^{\infty}$, defined by $r_k := \frac{\sum_{i=k+1}^{\infty} \lambda_i}{\lambda_{k+1}}$. We will use this, along with directly bounding $\mathcal{L}_{i,0}$.

If $K$ is a dot-product kernel on the sphere (such as the Gaussian kernel), we can take the eigenfunctions $\phi_i$ to be the spherical harmonics $Y_{ks}$, where $k \geq 0$ and $s \in [1, N(d, k)]$. Here $N(d, k) = \frac{(2k+d-2)(k+d-3)!}{k!(d-2)!}$ is the multiplicity of the $k$-th spherical harmonic. All $Y_{ks}$ for the same $k$ will have the same eigenvalue, which we will denote $\tilde{\lambda}_k$. In this case, we can write a closed-form expression for the eigenvalues of Gaussian kernel, $\tilde{\lambda}_k$, in terms of the bandwidth $\tau_m$ and modified Bessel functions of the first kind $I_v(x)$ [46] (see Appendix D). Using the closed form of $\tilde{\lambda}_i$ and the multiplicities of eigenvalues, we can understand how the test risk of the minimum norm interpolating solution $R(\hat{f}_0)$ behaves as $m \to \infty$, which tells us its overfitting behavior.

Additionally, $\mathcal{E}_0$ appearing in Equation (2), will be informative. For example, if $\lim_{m \to \infty} \mathcal{E}_0 > 1$ then the overfitting cannot be benign. The following bound using the effective rank $r_k$ holds [66]: For $k < m$ such that $r_k + k > m$ we have

$$\mathcal{E}_0 \leq \left(1 - \frac{k}{m}\right)^{-1} \left(1 - \frac{m}{k + r_k}\right)^{-1}. \tag{5}$$

For $k \geq m$ it holds that

$$\mathcal{E}_0 \geq \frac{1}{1 - \frac{m}{k}\left(\frac{k-m}{k-m+r_k}\right)}. \tag{6}$$

**Proof sketch of Theorem 3.** We will focus on the lower bounds in this case, as the result is negative. The key elements to understanding the effective rank $r_k$ and the test risk Equation (2) is to understand how the ratios of eigenvalues $\frac{\tilde{\lambda}_{k+1}}{\tilde{\lambda}_k}$ and the multiplicities $N(d, k)$ behave. Using the closed form of the eigenvalues of Gaussian kernel and the properties of modified Bessel functions [46, 48], with some computation (see Theorem 28 in the appendix for the computations), we can derive the following bounds on ratios of eigenvalues

$$\frac{\left(\frac{2}{\tau_m^2}\right)}{2\left(k + \frac{d}{2}\right) + \left(\frac{2}{\tau_m^2}\right)} < \frac{\tilde{\lambda}_{k+1}}{\tilde{\lambda}_k} < \frac{\left(\frac{2}{\tau_m^2}\right)}{\left(k + \frac{d}{2} - \frac{1}{2}\right) + \left(\frac{2}{\tau_m^2}\right)}. \tag{7}$$

From these bounds, we can derive tight bounds on $\frac{\tilde{\lambda}_{k+j}}{\tilde{\lambda}_k}$ for indices $k$ and $j$ using simple but long calculations (see Theorem 28 in the appendix). If $k = o\left(\frac{1}{\tau_m}\right)$ and $j = o\left(\frac{1}{\tau_m}\right)$, then $\frac{\tilde{\lambda}_{k+j}}{\tilde{\lambda}_k} \approx 1$. If $k \leq \Theta\left(\frac{1}{\tau_m}\right)$ and $j = \Theta\left(\frac{1}{\tau_m}\right)$, then $\frac{\tilde{\lambda}_{k+j}}{\tilde{\lambda}_k} = \Theta(1)$. If $k = \omega\left(\frac{1}{\tau_m}\right)$, then $\frac{\tilde{\lambda}_{k+j}}{\tilde{\lambda}_k} = o\left(\frac{1}{j^n}\right)$ for any integer $n \in \mathbb{N}$ (i.e. it deceases super-polynomialy). For $N(d, l)$ it holds that $N(d, l) = \Theta(l^{d-2})$ and $N_l := N(d, 1) + \cdots + N(d, l) = \Theta(l^{d-1})$. Therefore if $\tilde{l}$ is the index such that $\tilde{\lambda}_{\tilde{l}} = \lambda_l$, we have that $\tilde{l} = \Theta(l^{\frac{1}{d-1}})$.

For $\tau_m \geq \omega(m^{-\frac{1}{d-1}})$, we will take $l = (1 + \frac{1}{\sqrt{m}})m$ and show that $r_l = o(m)$. Then, the bound in Equation (6) will imply that $\mathcal{E}_0 \geq 1 + \sqrt{m}$, so from Equation (2), $\tilde{R}(\hat{f}_0) > \mathcal{E}_0 \sigma^2 = \sqrt{m}\sigma^2$. Note that for $l = (1 + \frac{1}{\sqrt{m}})m$, we have that $\tilde{l} = \Theta(m^{\frac{1}{d-1}}) = \omega\left(\frac{1}{\tau_m}\right)$. Note that for $r_{l-1}$, we have that

$$r_{l-1} = \sum_{i=0}^{\infty} \frac{\lambda_{l+i}}{\lambda_l} < N(d, \tilde{l}) + \sum_{i=1}^{\infty} N(d, \tilde{l} + i) \frac{\tilde{\lambda}_{\tilde{l}+i}}{\tilde{\lambda}_{\tilde{l}}} < \Theta(m^{\frac{d-2}{d-1}})(1 + \Theta(1)),$$

since $N(d, \tilde{l}+i) < N(d,j)i^{d-2}$ and $\frac{\tilde{\lambda}_{\tilde{l}+i}}{\tilde{\lambda}_{\tilde{l}}} < \frac{1}{i^d}$. So indeed $r_{l-1} = o(m)$ which implies $r_l = o(m)$.

For $\tau_m = o(m^{-\frac{1}{d-1}})$ and $\tau_m = \Theta(m^{-\frac{1}{d-1}})$, we will directly analyze Equation (2). For $k = \Theta(\frac{1}{\tau_m})$, we have that $\frac{\tilde{\lambda}_i}{\tilde{\lambda}_1} > \frac{1}{2}$ for all $i \leq k$. Let $\mathcal{L}_i = \frac{\tilde{\lambda}_i}{\tilde{\lambda}_i + \kappa_0}$. Note that $\mathcal{L}_i > \frac{1}{2}\mathcal{L}_1$ for $i \leq k$. Note that

$$ m = \sum_i N(d,i)\mathcal{L}_i > \frac{1}{2}\mathcal{L}_1 \left( N(d,1) + \cdots + N(d,k) \right) > \Theta\left( \left( \frac{1}{\tau_m} \right)^{d-1} \right) \mathcal{L}_1. $$

So, we have that for all $i$ that $\mathcal{L}_i < \mathcal{L}_1 < \frac{m}{\Theta\left( \left( \frac{1}{\tau_m} \right)^{d-1} \right)}$. From Equation (2), we have that

$$ \tilde{R}(\hat{f}_0) = \mathcal{E}_0 \sum_i N(d,i)(1-\mathcal{L}_i)^2 \beta_i^2 + \mathcal{E}_0 \sigma^2 > \mathcal{E}_0 \left( 1 - \frac{m}{\Theta\left( \left( \frac{1}{\tau_m} \right)^{d-1} \right)} \right)^2 \|f^*\|^2 + \mathcal{E}_0 \sigma^2. $$

For $\tau_m = o(m^{\frac{1}{d-1}})$ this is sufficient. Additionally, by a similar computation as above, in this case, $r_1 = \omega(m)$, so $\mathcal{E}_0$ is bounded by Equation (5). For $\tau_m = \Theta(m^{\frac{1}{d-1}})$, using Equation (5) and Equation (6), by showing that for $l = \Theta(m)$, $r_l = \Theta(m)$, we have $\Theta(1) > \mathcal{E}_0 > 1 + \Theta(1)$.

**Proof sketch of Theorem 7.** Let $\mathcal{L}_i = \frac{\tilde{\lambda}_i}{\tilde{\lambda}_i + \kappa_0}$. Note that $(1-\mathcal{L}_i)^2 = \frac{\kappa_0^2}{(\kappa_0 + \tilde{\lambda}_i)^2}$. Then, Equation (2) can be rewritten and bounded as (with an abuse of notation for $\beta_i$)

$$ \tilde{R}(\hat{f}_0) = \mathcal{E}_0 \sum_i N(d,i)(1-\mathcal{L}_i)^2 \beta_i^2 + \mathcal{E}_0 \sigma^2 \leq \mathcal{E}_0 B^2 \sum_{i=1}^{l} N(d,i) \frac{\kappa_0^2}{(\kappa_0 + \tilde{\lambda}_i)^2} + \mathcal{E}_0 \sigma^2. $$

Note that $\frac{\kappa_0^2}{(\kappa_0 + \tilde{\lambda}_i)^2} < \frac{\kappa_0^2}{\tilde{\lambda}_i^2}$ and also $\mathcal{L}_i \leq \frac{\tilde{\lambda}_i}{\kappa_0}$. Therefore, we have that $\tilde{R}(\hat{f}_0) \leq \mathcal{E}_0 B^2 \kappa_0^2 \left( \sum_{i=1}^{l} N(d,i)\frac{1}{\tilde{\lambda}_i^2} \right) + \mathcal{E}_0 \sigma^2$. Note that $m = \sum_i N(d,i)\mathcal{L}_i < \sum_i N(d,i)\frac{\tilde{\lambda}_i}{\kappa_0} < \frac{A}{\kappa_0}$, so $\kappa_0 < \frac{A}{m}$. Finally, to bound $\mathcal{E}_0$, note that in Equation (5) we can choose $k = L_m < m$, then $r_k + k > U_m > m$, so $\mathcal{E}_0 \leq \left( 1 - \frac{L_m}{m} \right)^{-1} \left( 1 - \frac{m}{U_m} \right)^{-1}$. Combining these with $\kappa_0 < \frac{A}{m}$ gives the claim of Theorem 7. The proof of the alternative bound is harder and will be delayed to the appendix.

**Proof sketch of Theorem 9.** By Theorem 3 from Zhou et al. [66], if $k$ is the first $k < m$ such that $k + br_k \geq m$, then $\mathcal{E}_0 \geq \left( 1 - \left( \frac{b}{b+1} \right)^2 \frac{k}{m} \right)^{-1}$. Since $\max_{i \leq k_m} \left( \frac{1}{\tilde{\lambda}_i} \right) < \frac{m - L_m}{b}$, for $l < N(d,1) + \cdots + N(d,k_m)$, we have that $br_l + l \leq N(d,1) + \cdots + N(d,k_m) - 1 + b\max_{i \leq k_m} \left( \frac{1}{\tilde{\lambda}_i} \right) < L_m + m - L_m \leq m$, so the first $l$ for which $r_l + l > m$ is $l = L_m = N(d,1) + \cdots + N(d,k_m)$. Plugging in $k = L_m$ we get that $\mathcal{E}_0 \geq \left( 1 - \left( \frac{b}{b+1} \right) \frac{L_m}{m} \right)^{-1}$, so from Equation (6), we have

$$ \tilde{R}(\hat{f}_0) \geq \left( 1 - \left( \frac{b}{b+1} \right)^2 \frac{L_m}{m} \right)^{-1} \sigma^2. $$

**Proof sketch of Corollary 11.** Note that an analogous proof holds for any $\frac{d^\alpha}{m} \to c$, for a constant $c$, but we take equality for simplicity. If $k$ is a constant, i.e. it does not change with the dimension, we have that $N(d,k) = \Theta(d^k)$. Therefore, $k_m = \lfloor \alpha \rfloor$ if $\alpha$ is a non-integer and $\alpha - 1$ if $\alpha$ is an integer. If $\alpha$ is a non-integer, then $L_m = N(d,k_m) + \cdots + N(d,1) = \Theta(d^{\lfloor \alpha \rfloor})$ and $U_m = N(d,k_m+1) + \cdots + N(d,1) = \Theta(d^{\lfloor \alpha \rfloor + 1})$. So we have that $\frac{L_m}{m} = d^{\lfloor \alpha \rfloor - \alpha}$, $\frac{m}{L_m} = d^{\alpha - \lfloor \alpha \rfloor - 1}$, $N(d, \lfloor \alpha \rfloor) = d^{\lfloor \alpha \rfloor}$. Note that $k_m = \lfloor \alpha \rfloor$. We have from Assumption 10 that $\max_{i \leq k_m} \frac{1}{\tilde{\lambda}_i} = O(N(d,k_m))$. Note that Theorem 7 holds with $\left( \sum_{i=1}^{l} N(d,i)\frac{1}{\tilde{\lambda}_i} \right)$ instead of $\left( \sum_{i=1}^{l} N(d,i)\frac{1}{\tilde{\lambda}_i^2} \right)$. Therefore, we have

$\frac{1}{m^2} \max_{l \leq k_m} \left( \frac{1}{\tilde{\lambda}_i} N(d, i) \right) \leq O(d^{(2\lfloor \alpha \rfloor - 2\alpha)})$. This gives

$$\tilde{R}(\hat{f}_0) \leq \left( 1 - d^{\lfloor \alpha \rfloor - \alpha} \right)^{-1} \left( 1 - d^{\alpha - \lfloor \alpha \rfloor - 1} \right)^{-1} \sigma^2$$
$$+ O \left( B^2 \left( 1 - d^{\lfloor \alpha \rfloor - \alpha} \right)^{-1} \left( 1 - d^{\alpha - \lfloor \alpha \rfloor - 1} \right)^{-1} d^{(2\lfloor \alpha \rfloor - 2\alpha)} \right).$$

Therefore, $\lim_{m \to \infty} \tilde{R}(\hat{f}_0) = \sigma^2$. So, if $\alpha$ is not an integer, we get benign overfitting. If $\alpha$ is an integer, note that $N(d, 1) + \cdots + N(d, \alpha - 1) = o(d^\alpha)$, so $k_m = \alpha$. Then there are $c_u, c_l \in (0, 1)$ such that $c_l < \frac{L_m}{m} \leq c_u < 1$. So, Theorem 9 holds for $b = \frac{1}{2}$, so $\tilde{R}(\hat{f}_0) > \frac{1}{1 - \frac{c_l}{9}} \sigma^2$. This shows that $\liminf_{m \to \infty} \tilde{R}(\hat{f}_0) \geq \frac{1}{1 - \frac{c_l}{9}} \sigma^2 > \sigma^2$. The upper bound follows as above. Since we assumed that $\max_{i \leq k_m} \frac{1}{\tilde{\lambda}_i} = O(N(d, k_m))$, and $\max_{i \leq k_m} N(d, i) = N(d, k_m) = \Theta(d^\alpha)$, we have that $\sum_{i=1}^{k_m} N(d, i) \frac{1}{\tilde{\lambda}_i} = \Theta(d^{2\alpha})$, so

$$\tilde{R}(\hat{f}_0) \leq (1 - c_l)^{-1} \left( 1 - d^{-1} \right)^{-1} \sigma^2 + \Theta \left( B^2 \left( 1 - d^{-1} \right)^{-1} \right).$$

So, we conclude that for integer $\alpha$ we have tempered overfitting and for non-integer $\alpha$ we have benign overfitting. This recovers results of Ghorbani et al. [24], Mei et al. [38], Misiakiewicz [40], Barzilai and Shamir [4], Zhang et al. [64].

**Proof sketch of Corollary 12.** Note that this holds for any scaling of $d$ and $m$ where $d = \Theta(\log m)$, but we take this particular one for concreteness. As we show in the appendix (Theorem 21), it is not hard to see that $L_m \approx \alpha_l m$ for some constant $\alpha_l < 1$ and $L_m \approx \alpha_u m$ for some constant $\alpha_u > 1$. Theorem 9 implies that we cannot get benign overfitting in this case, i.e. for all $m$, $\tilde{R}(\hat{f}_0) > \frac{1}{1 - \left( \frac{b}{b+1} \right)^2 \alpha_l} \sigma^2$. This shows that $\liminf_{m \to \infty} \tilde{R}(\hat{f}_0) \geq \frac{1}{1 - \left( \frac{b}{b+1} \right)^2 \alpha_l} \sigma^2 > \sigma^2$.

**Proof sketch of Corollary 13.** We have that for the Gaussian kernel on the sphere, Theorem 7 holds with $A = 1$ (see Appendix C.4 for further details). First, we will compute $k_m$ and hence $L_m$ and $U_m$. After some tedious calculation (Theorem 23 in the appendix), we see that for $k \leq 2^l + l - 1$ we have $N(d, 1) + \cdots + N(d, k) = o(m)$, but for $k = 2^l + l$ we have $N(d, 1) + \cdots + N(d, k) > m$. This shows that $k_m = 2^l + l - 1$ and so again after long calculations $L_m = \Theta(\frac{m}{\log m})$ and $U_m > \Theta(md^{0.89})$. To estimate $\sum_{i=1}^{k} N(d, i) \tilde{\lambda}_i$, note that eigenvalues are decreasing and $iN(d, i) = iN(d, i+1) \frac{2i+d-2}{2i+d} \frac{i+1}{i+d-2} = o(N(d, i+1))$ (take $k \ll d$), so it suffices to estimate $N(d, k) \frac{1}{\tilde{\lambda}_k}$. Now, from Equation (7) and an estimate on the size of the first eigenvalue (Corollary 31) $\tilde{\lambda}_1 = \Theta(\frac{1}{d^\alpha})$, for fixed $\alpha > 0$, we can estimate the size $\tilde{\lambda}_k$. We have that

$$\frac{1}{\tilde{\lambda}_k} < \frac{1}{\tilde{\lambda}_1} (\tau_m)^k \left( \frac{d}{2} + k + \frac{2}{\tau_m^2} \right)^k < d^{k(1+\varepsilon)},$$

for arbitrarily small $\varepsilon$. For $k = \frac{7k_m}{24}$, from additional calculation (Theorem 23 in the appendix), we have that $m^{\frac{1}{4}} \leq N(d, k) \leq m^{\frac{1}{3}}$ and $d^k < m^{\frac{7}{24}}$. So $\sum_{i=1}^{k_m} N(d, i) \frac{1}{\tilde{\lambda}_i^2} < m$. Plugging these back into Theorem 7 gives the desired result.

## 6  Summary

In this paper, we considered the minimum norm interpolating solution of kernel ridge regression in fixed dimension with Gaussian kernel and varying or adaptively chosen bandwidth, and in increasing dimension with various kernels. In fixed dimension, we showed that if the source distribution is uniform on the sphere, then the minimum norm interpolating solution is inconsistent for any choice of bandwidth, and usually worse than the null predictor, except possibly in one particular scaling of bandwidth and with small noise. Furthermore, we showed a general upper and lower bound on the test risk, which we applied in the case of increasing dimension to recover the currently known results about polynomially increasing dimension and show two new results: We showed that no dot-product kernel on the sphere can achieve consistency for logarithmic scaling of the dimension, and obtained the first case of sub-polynomially scaling dimension where the minimum norm interpolating solution exhibits benign overfitting, namely with Gaussian kernel and dimension scaling as $d = \exp(\sqrt{\log m})$.

## Acknowledgments and Disclosure of Funding

We would like to thank Theodor Misiakiewicz and Sam Buchanan for useful discussions.

This work was done as part of the NSF-Simons funded collaboration on the Theoretical Foundations of Deep Learning (https://deepfoundations.ai), and primarily while GV was at TTIC. GV is supported by research grants from the Center for New Scientists at the Weizmann Institute of Science, and the Shimon and Golde Picker – Weizmann Annual Grant.

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

# A    Empirical Justification

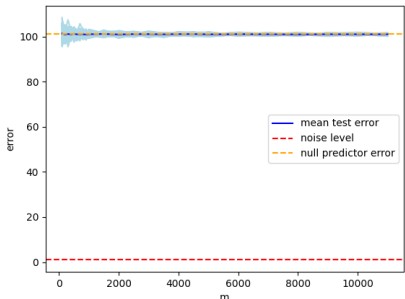

(a) $\tau_m = o(m^{-\frac{1}{d-1}})$ with $\sigma^2 = 1$ and $d = 6$. We see that the test error tends to something that is equal to or larger than the test risk of the null predictor.

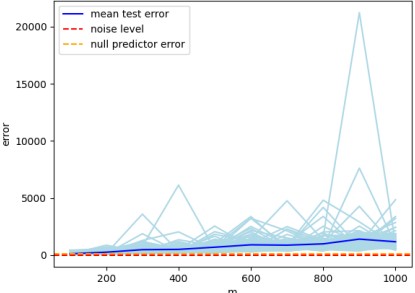

(b) $\tau_m = \omega(m^{-\frac{1}{d-1}})$ with $\sigma^2 = 10$ and $d = 4$. We see that the test error is increasing with the sample size $m$, suggesting that the test error diverges to infinity.

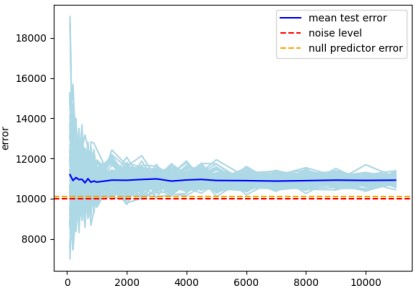

(c) $\tau_m = \Theta(m^{-\frac{1}{d-1}})$ with $\sigma^2 = 10000$ and $d = 6$. We see that the test error is well above the risk of the null predictor.

Figure 1: We plot the dependence of the test error of the Gaussian kernel ridgeless predictor for $y = f^*(x) + \xi$, with $\xi \sim N(0, \sigma^2)$, $f^* = 10$, $x \sim \mathrm{Unif}(S^{d-1})$, and $\sigma^2$ and $d$ as above, and sample size $m$. Light blue lines represent the test error for $100$ different runs of the experiment and dark blue is the mean of all those runs. The red dotted line is the noise level $\sigma^2$ and the yellow dotted line is the error the null predictor achieves in this setting. We see that the test error concentrates around its mean and that the behavior is as predicted by Theorem 3.

In this section, we provide empirical justification for the predictions of Theorem 3, which deals with the case of fixed dimension. Figure 1 shows the dependence of the test error on the sample size for the Gaussian Kernel Ridgeless prediction with varying bandwidth, as in the setup of Theorem 3. Specifically, we consider $y = f^*(x) + \xi$ where $\xi \sim N(0, \sigma^2)$, $f^* = 10$, $\sigma^2$ is the noise level, and

$x \sim \text{Unif}(\mathbb{S}^{d-1})$. We vary the values of $d$ and $\sigma^2$ and the bandwidth scaling $\tau_m$ as follows: for $\tau_m = o(m^{-\frac{1}{d-1}})$ we take $\sigma^2 = 1$ and $d = 6$, for $\tau_m = \omega(m^{-\frac{1}{d-1}})$ we take $\sigma^2 = 10$ and $d = 4$, and for $\tau_m = \Theta(m^{-\frac{1}{d-1}})$ we take $\sigma^2 = 10000$ and $d = 6$. We compare the test error (light blue) and the mean of the test error for 100 different runs of the experiment (dark blue) with the noise level (the Bayes risk; red) and the risk of the null predictor (yellow). We see that for all three cases, our predictions agree with the experiments, that is, in the case $\tau_m = o(m^{-\frac{1}{d-1}})$ the test error is eventually above the null predictor error but finite, in the case $\tau_m = \omega(m^{-\frac{1}{d-1}})$ the test error increases with sample size, and in the case $\tau_m = \Theta(m^{-\frac{1}{d-1}})$ the test error is above null predictor error. The code to reproduce these experiments can be found at

https://github.com/marko-medvedev/overfitting-behavior-of-gaussian-kernel-ridgeless-regression.

We ran the experiments on one A6000 GPU.

**Statistical significance of experimental results:** In Figure 2 and Figure 3, we present evidence for the statistical significance of our experimental results. We examine whether a significant mass of the distribution of the mean of the test loss, $L$, exhibits the same behavior as $L$. We use 50 of the 100 experiments to estimate an interval $[L_{\text{low}}, L_{\text{high}}]$ and the other 50 to test whether this interval contains $p = 0.8$ of the total mass of the distribution of $L$ at $\alpha = 0.05$ significance level. In Figure 2, we plot $L_{\text{low}}$ and $L_{\text{high}}$, the dark blue lines. We perform one lower-tailed and one upper-tailed test [17]. The null hypotheses are: $1 - \frac{1-p}{2}$ population quantile is no greater than $L_{\text{high}}$ and $\frac{1-p}{2}$ population quantile is at least as great as $L_{\text{low}}$. Indeed, both $L_{\text{low}}$ and $L_{\text{high}}$ have the desired behavior. Furthermore, in Figure 3, we report the relevant test statistics, $T_1$ which is the number of draws of $L$ no greater than $L_{\text{high}}$ and $T_2$ which is the number of draws of $L$ less than $L_{\text{low}}$. $t_1$ and $t_2$ are the minimum numbers such that $P(Y \leq t_1) = \alpha$ and $P(Y \leq t_2) = 1 - \alpha$, where $Y$ is a binomially distributed random variables with parameters $p$ and $n = 50$. The leftmost bar plot compares $T_1$ and $t_1$, the middle bar plot compares $T_2$ and $t_2$, and the rightmost bar plot shows the $p$-value and the significance $\alpha$. To accept the null hypothesis we need $T_1 > t_1$, $T_2 \leq t_2$, and high $p$-value. We report an aggregated p-value. We accept the null hypothesis in both cases - the test statistics $T_1$ and $T_2$ satisfy the required conditions and the $p$-values are high.

Overall, we can conclude that the prediction of Theorem 3 agrees with the empirical observations that are statistically significant.

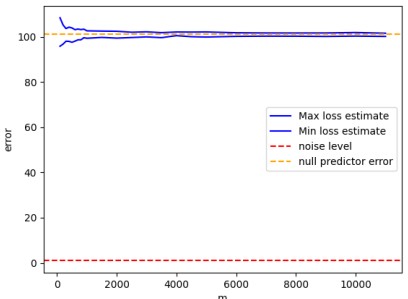

(a) $\tau_m = o(m^{-\frac{1}{d-1}})$ with $\sigma^2 = 1$ and $d = 6$.

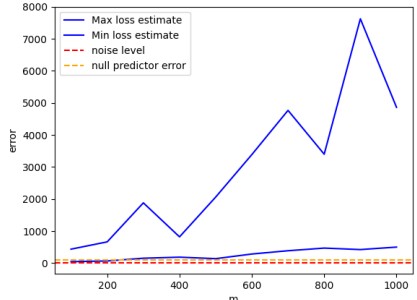

(b) $\tau_m = \omega(m^{-\frac{1}{d-1}})$ with $\sigma^2 = 10$ and $d = 4$.

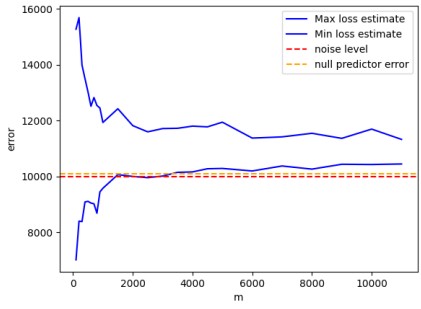

(c) $\tau_m = \Theta(m^{-\frac{1}{d-1}})$ with $\sigma^2 = 10000$ and $d = 6$.

Figure 2: For each of the setups of Theorem 3, we plot the dependence of the estimates for the minimum and maximum of the test error of the Gaussian kernel ridgeless predictor $L_{\text{low}}$ and $L_{\text{high}}$ for $y = f^*(x) + \xi$, with $\xi \sim N(0, \sigma^2)$, $f^* = 10$, $x \sim \text{Unif}(S^{d-1})$, and $\sigma^2$ and $d$ as above, on the sample size $m$. The red dotted line is the noise level $\sigma^2$ and the yellow dotted line is the error the null predictor achieves in this setting. In dark blue, we plot the estimates for the minimum and maximum of mean test error $L_{\text{low}}$ and $L_{\text{high}}$ for a given sample size. We test whether $[L_{\text{low}}, L_{\text{high}}]$ contains $p = 0.8$ of the total mass of the mean test error $L$.

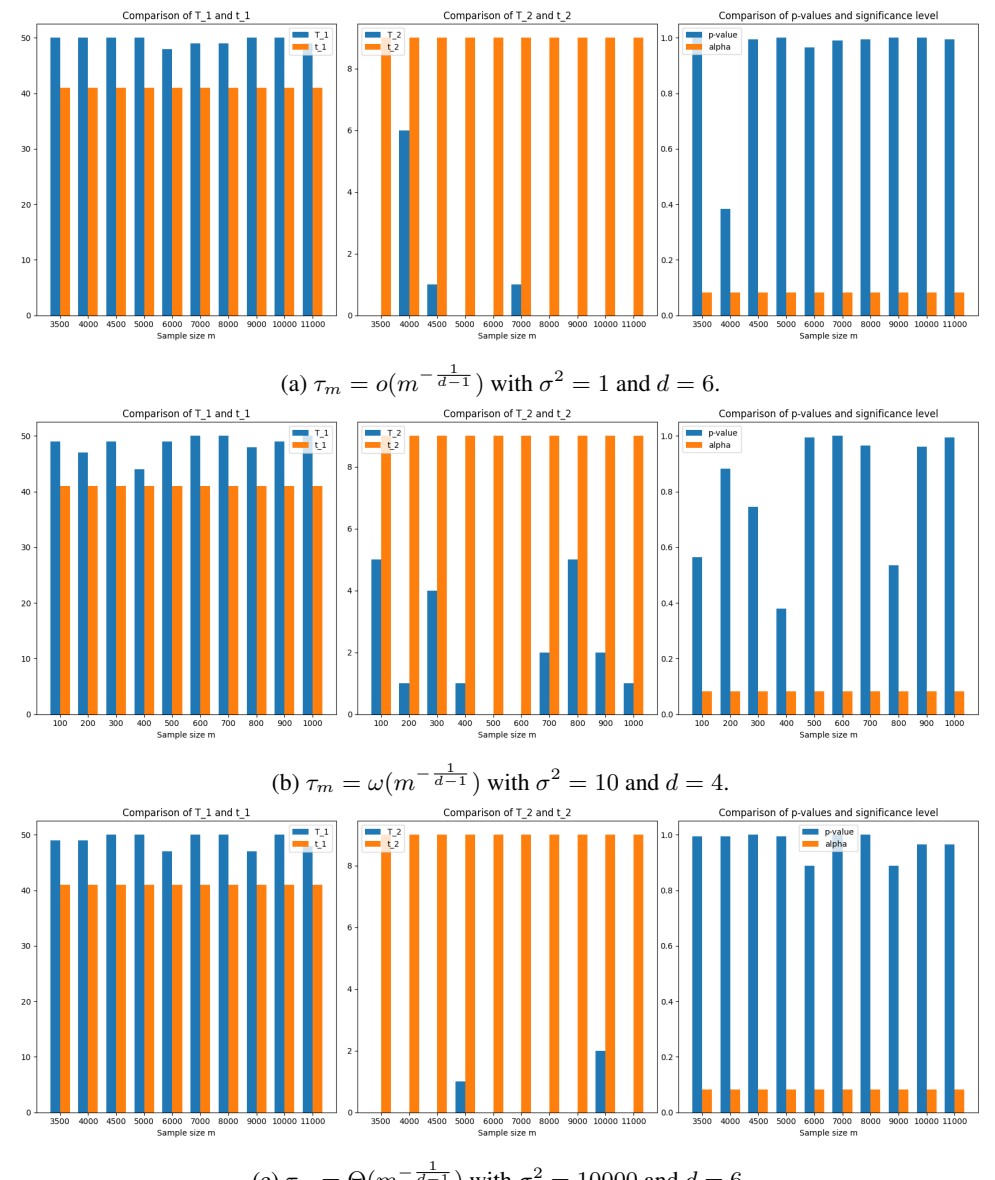

(a) $\tau_m = o(m^{-\frac{1}{d-1}})$ with $\sigma^2 = 1$ and $d = 6$.

(b) $\tau_m = \omega(m^{-\frac{1}{d-1}})$ with $\sigma^2 = 10$ and $d = 4$.

(c) $\tau_m = \Theta(m^{-\frac{1}{d-1}})$ with $\sigma^2 = 10000$ and $d = 6$.

Figure 3: For each of the setups of Theorem 3, we test the statistical significance that $p = 0.8$ of the probability mass of $L$, the test error, is inside $[L_{\text{low}}, L_{\text{high}}]$ at $\alpha = 0.05$ significance level. In the leftmost plots, we plot the dependence of the test statistics $T_1$ and $t_1$ on sample size $m$. $T_1$ is the number of draws of the test error $L$ no greater than $L_{\text{high}}$ and $t_1$ is the minimal number such that $P(Y \leq t_1) = \alpha$. In the middle plots, we plot the dependence of test statistics $T_2$ and $t_2$ on sample size $m$. $T_2$ is the number of draws of the test error $L$ less than $L_{\text{low}}$ and $t_2$ is the minimal number such that $P(Y \leq t_2) = 1 - \alpha$. Here $Y$ is a binomially distributed random variable with parameters $p$ and $n = 50$. On the rightmost plots, we show the dependence of the aggregated $p$-value on the sample size $m$. We accept the null hypothesis in both cases because $T_1 > t_1$, $T_2 \leq t_2$, and the $p$-values are high.

# B The case of fixed dimension

**Cost of overfitting** The cost of overfitting, introduced in [66], is defined as the ratio of the test risk of an interpolating solution and the test risk of the optimally regularized solution. It measures how worse off are we interpolating rather than regularizing.

**Definition 15** (Cost of overfitting). Let $\hat{f}_{\delta^*}$ be the optimally regularized solution, i.e.

$$R(\hat{f}_{\delta^*}) = \inf_{\delta \geq 0} R(\hat{f}_\delta).$$

For a distribution $\mathcal{D}$ over $\mathcal{X} \times \mathbb{R}$, we will define the cost of overfitting $C(\mathcal{D}, m)$ as

$$C(\mathcal{D}, m) = \frac{R(\hat{f}_0)}{R(\hat{f}_{\delta^*})}.$$

Zhou et al. [66] show that agnostically to the distribution $\mathcal{D}$, the behavior of the cost of overfitting is tightly controlled by the effective ranks of systems of eigenvalue $\{\lambda_i\}_{i=1}^\infty$ of a given kernel (coming from Mercer's theorem)

$$r_k := \frac{\sum_{i=k+1}^\infty \lambda_i}{\lambda_{k+1}} \quad \text{and} \quad R_k := \frac{\left(\sum_{i=k+1}^\infty \lambda_i\right)^2}{\sum_{i=k+1}^\infty \lambda_i^2}.$$

We will assume additionally that the eigenvalues are nonzero, so $r_k$ and $R_k$ are well defined on $\mathbb{N}$. Since we chose that the eigenvalues are sorted and since they square summable, we have that

$$r_k \leq R_k \leq r_k^2.$$

The following two results of [66] summarize how we will use effective ranks to control the cost of overfitting. Proposition 16 gives a necessary and sufficient condition for the overfitting to benign in terms of the cost of overfitting, i.e. that $C(\mathcal{D}, m) \to 1$, in terms of effective ranks.

**Proposition 16** (Necessary and sufficient condition for benign overfitting [66]). Let $k_n$ be the smallest integer $k \in \mathbb{N} \cup \{0\}$ for which $n < k + r_k$. Then $\mathcal{E}_0 \to 1$ if and only if

$$\lim_{n \to \infty} \frac{k_n}{n} = 0 \quad \text{and} \quad \lim_{n \to \infty} \frac{n}{R_{k_n}} = 0.$$

Proposition 17 gives a sufficient condition for $\tilde{R}(\hat{f}_0) \to \infty$ in terms of effective ranks.

**Proposition 17** (Sufficient condition for catastrophic overfitting [66]). Let $\varepsilon > 0$ and let $k = (1+\varepsilon)m$. If $\lim_{k \to \infty} \frac{r_k}{k} = 0$ then $\mathcal{E}_0 \to \infty$, i.e. the overfitting is catastrophic.

Using sharp bounds on eigenvalues $\{\lambda_i\}_{i=1}^\infty$, derived in Theorem 28, we can derive bounds on effective ranks so that we can apply the results of [66] and Equation (2) to bound the test risk of minimum norm interpolant.

When the input is distributed uniformly on the sphere, the eigenfunctions $\{\phi_k\}$ in the Mercer decomposition of the kernel function can be taken to be the spherical harmonics $Y_{ks}$. Using the Funk-Hecke formula we can find the closed form of eigenvalues for the Gaussian kernel [46], which are given in terms of bandwidth $\tau_m$ and modified Bessel function of the first kind $I_v(x)$. In Appendix D, we will derive bounds on the sizes of the eigenvalues using the properties of Bessel functions and their multiplicities.

We need to take a particular input distribution, in order to able to compute the eigenvalues explicitly and to know their multiplicities. The result of Cao et al. [9] offers one possible way of generalizing our result to more general different distributions

We split the proof into two parts. First, we will bound the cost of overfitting. Then, we will provide lower bounds on the test risk of Gaussian KRR.

**Bounds on cost of overfitting** The bounds on cost of overfitting will be used to show whether the test risk is bounded above or away from Bayes risk.

**Lemma 18** (Cost of overfitting for Gaussian KRR). Let $\mathcal{X} \sim \text{Unif}(\mathcal{S}^{d-1})$. Then the following bounds hold for the minimum norm interpolating solution of KRR and its cost of overfitting $C(\mathcal{D}, m)$ and $\mathcal{E}_0$:

1. If $\tau_m \leq m^{-\frac{1}{d-1}} t(m)$, where $t(m) \to 0$ as $m \to \infty$ then there is a constant $C_I$ that depends on $d$ and $C$, such that

$$1 \leq C(\mathcal{D}, m) \leq \mathcal{E}_0 \leq \left(1 - \frac{1}{m}\right)^{-2} \left(1 - C_I t(m)^{\frac{d-1}{2}}\right)^{-1}.$$

2. If $\tau_m \geq m^{-\frac{1}{d-1}} t(m)$, where $t(m) \to \infty$ as $m \to \infty$ then there is a integer $m_0$ depending on $d$ and $t(m)$ such that for all $m \geq m_0$

$$\mathcal{E}_0 \geq \sqrt{m}.$$

3. If $C_1 m^{-\frac{1}{d-1}} \leq \tau_m \leq C_2 m^{-\frac{1}{d-1}}$ then there exist constants $C_I > 0$ and $C_{II}$ depending on $C_1, C_2$ and $d$ such that

$$C(\mathcal{D}, m) \leq \mathcal{E}_0 \leq C_I$$
$$\mathcal{E}_0 \geq 1 + C_{II}.$$

*Proof.* We will break down the proof into the proof of the three parts.

**Part 1:** We will use Proposition 16 to prove this that $\mathcal{E}_0 \to 1$. We can see the upper bound on $\mathcal{E}_0$ from Theorem 2 [66]. We have that for $k < m$

$$\mathcal{E}_0 \leq \left(1 - \frac{k}{m}\right)^{-2} \left(1 - \frac{m}{R_k}\right)^{-1}.$$

Now we will show that for $k = 1$, we have that $k + r_k > m$ and $\lim_{m \to \infty} \frac{m}{R_1} = 0$. This implies that conditions of Proposition 16 hold so $\mathcal{E}_0 \to 1$. We will rewrite $r_k = \frac{\sum_{i>k} \lambda_k}{\lambda_{k+1}}$ using $\tilde{\lambda}_i$. We have that

$$r_1 = \frac{\sum_{i>1} \lambda_i}{\lambda_2} = \tilde{N}(d, \tilde{1}) + \sum_{j=1}^{\infty} N(d, \tilde{1} + j) \frac{\tilde{\lambda}_{\tilde{1}+j}}{\tilde{\lambda}_{i_1}},$$

where $N(d, 1) = d$, so $\tilde{1} = 1$ and $\tilde{N}(d, \tilde{1}) = d - 1$. Therefore, we have

$$r_1 = (d-1) + \sum_{j=1}^{\infty} N(d, 1+j) \frac{\tilde{\lambda}_{1+j}}{\tilde{\lambda}_1}.$$

Let $T = \frac{2}{\tau_m^2}$. Taking $\frac{\sqrt{T}g(m)}{2} \leq j \leq \sqrt{T}g(m)$, $k = 1$, we can apply the bound Theorem 28 on $\frac{\tilde{\lambda}_{1+j}}{\tilde{\lambda}_1}$. Therefore, we have that

$$\exp(-5g(m)) < \frac{\tilde{\lambda}_{1+j}}{\tilde{\lambda}_1} < 1.$$

In particular, we have that for $m$ large enough, for all such $j$ it holds that $\frac{\tilde{\lambda}_{1+j}}{\tilde{\lambda}_1} > \frac{1}{2}$. Going back to $r_1$, this means that

$$r_1 = (d-1) + \sum_{j=1}^{\infty} N(d, 1+j) \frac{\tilde{\lambda}_{1+j}}{\tilde{\lambda}_1}$$

$$> \sum_{j=\frac{1}{2}\sqrt{T}g(m)}^{\sqrt{T}g(m)} N(d, 1+j) \frac{\tilde{\lambda}_{1+j}}{\tilde{\lambda}_1} > \sum_{j=\frac{1}{2}\sqrt{T}g(m)}^{\sqrt{T}g(m)} (1+j)^{d-2} \frac{1}{2}$$

$$> \sum_{j=\frac{1}{2}\sqrt{T}g(m)}^{\sqrt{T}g(m)} \frac{1}{2^{d+1}} (\sqrt{T}g(m))^{d-2} > \frac{1}{2}\sqrt{T}g(m) \frac{1}{2^{d+1}} (\sqrt{T}g(m))^{d-2}$$

$$> \frac{1}{2} (\sqrt{T}g(m))^{d-1}.$$

If $\sqrt{T}g(m)$ is not an integer, we can take its integer part and add a small constant since $\sqrt{T}g(m)$ will be large. Note that $\tau_m \leq m^{-\frac{1}{d-1}} t(m)$, so

$$T \geq m^{\frac{2}{d-1}} \frac{1}{t(m)^2}.$$

Therefore

$$r_1 > \frac{1}{2}(\sqrt{T}g(m))^{d-1} > \frac{1}{2}\left(m^{\frac{1}{d-1}}\frac{g(m)}{t(m)}\right)^{d-1} = m^1 \left(\frac{g(m)}{t(m)}\right)^{d-1},$$

so as long as $\frac{g(m)}{t(m)} \to \infty$, $r_1 > m$ for $m$ large enough. We can take $g(m) = \sqrt{t(m)}$. Then, in Proposition 16, we can select $k_m = 1$, since $1 + r_1 > m$ for large enough $m$. Note also that $R_1 \geq r_1$, so we also have that $R_1 \geq r_1 > m^1 \left(\frac{g(m)}{t(m)}\right)^{d-1}$, so

$$\lim_{m\to\infty} \frac{1}{m} = 0 \text{ and } \lim_{m\to\infty} \frac{m}{R_1} = 0.$$

This finishes the proof of the first claim.

**Part 2:** Let $\tau_m \geq m^{-\frac{1}{d-1}}t(m)$, where $t(m) \to \infty$ as $m \to \infty$. We will use Proposition 17 to prove this claim. Using Theorem 5 from [66] we know that for any $\varepsilon > 0$ if $r_k = o(m)$ for $k = (1+\varepsilon)m$ then the following bound holds

$$\mathcal{E}_0 \geq \frac{1}{1 - \frac{m}{k}\left(\frac{k-m}{k-m+r_k}\right)} > 1 + \frac{1}{\varepsilon}.$$

We will show that there is a constant $U > 0$ such that for any $k \geq m$, we have that

$$r_k \leq Um\left(\frac{1}{t(m)}\right)^{\frac{d-1}{2}}.$$

By Theorem 28, we have that for $j \geq \sqrt{T}g(m)$ where $g(m) \to \infty$ as $m \to \infty$ that

$$\frac{\tilde{\lambda}_{\tilde{k}+j}}{\tilde{k}} < \exp\left(-\frac{g(m)^2}{4}\right).$$

Let $k = (1+\varepsilon)m$. We have that

$$r_k = \tilde{N}(d,\tilde{k}) + \sum_{j=1}^{\infty} N(d,\tilde{k}+j)\frac{\tilde{\lambda}_{\tilde{k}+j}}{\tilde{\lambda}_{\tilde{k}}}.$$

Now, if we take $j \geq j_0 = \sqrt{T}g(m)$, it holds that

$$\frac{\tilde{\lambda}_{\tilde{k}+j}}{\tilde{\lambda}_{\tilde{k}}} < \exp(-\frac{g(m)^2}{4}).$$

Therefore, we have that

$$\begin{aligned}
r_k &= \tilde{N}(d,\tilde{k}) + \sum_{j=1}^{\infty} N(d,\tilde{k}+j)\frac{\tilde{\lambda}_{\tilde{k}+j}}{\tilde{\lambda}_{\tilde{k}}} \\
&= \tilde{N}(d,\tilde{k}) + \sum_{j=1}^{j-1} N(d,\tilde{k}+j)\frac{\tilde{\lambda}_{\tilde{k}+j}}{\tilde{\lambda}_{\tilde{k}}} + \sum_{j=j}^{\infty} N(d,\tilde{k}+j)\frac{\tilde{\lambda}_{\tilde{k}+j}}{\tilde{\lambda}_{\tilde{k}}} \\
&\leq n_u j^{d-1} + \sum_{j=j}^{\infty} N(d,\tilde{k}+j)\frac{\tilde{\lambda}_{\tilde{k}+j}}{\tilde{\lambda}_{\tilde{k}}}.
\end{aligned}$$

Note that by Theorem 28, for $j \geq l\sqrt{T}g(m)$ we have that

$$\frac{\tilde{\lambda}_{\tilde{k}+j}}{\tilde{k}} < \exp\left(-\frac{l^2 g(m)^2}{4}\right).$$

Note also that this bounds holds for all $l$ simultaneously. Therefore, we can write

$$r_k \leq n_u j^{d-1} + \sum_{j=j}^{\infty} N(d, \tilde{k}+j) \frac{\tilde{\lambda}_{\tilde{k}+j}}{\tilde{\lambda}_{\tilde{k}}}$$

$$= n_u j^{d-1} + \sum_{l=1}^{\infty} \sum_{s=1}^{j} N(d, lj+s) \frac{\tilde{\lambda}_{\tilde{k}+lj+s}}{\tilde{\lambda}_{\tilde{k}}}$$

$$\leq n_u j^{d-1} + \sum_{l=1}^{\infty} \sum_{s=1}^{j} N(d, (l+1)j) \frac{\tilde{\lambda}_{\tilde{k}+lj}}{\tilde{\lambda}_{\tilde{k}}}$$

$$\leq n_u j^{d-1} + \sum_{l=1}^{\infty} (l+1)^{d-1} j^{d-1} \exp(-\frac{l^2 g(m)^2}{4})$$

$$\leq m \left( \frac{g(m)}{t(m)} \right)^{d-1} \left( n_u + \sum_{l=1}^{\infty} (l+1)^{d-1} \exp(-\frac{l^2}{4}) \right)$$

$$\leq U m \left( \frac{g(m)}{t(m)} \right)^{d-1} = o(m).$$

Second to last inequality is true because $g(m) \to \infty$ as $m \to \infty$. The last inequality follows from the fact that $\sum_{l=1}^{\infty} (l+1)^{d-1} \exp(-\frac{l^2}{4})$ is bounded by a constant. Note that the bound is independent of $k$, so we can vary $\varepsilon$ with $m$ as well. If we choose $\varepsilon = \frac{1}{\sqrt{m}}$, the desired result follows.

**Part 3:**  Let $C_1 m^{-\frac{1}{d-1}} \leq \tau_m \leq C_2 m^{-\frac{1}{d-1}}$. Note that the following two bounds hold for $\mathcal{E}_0$ [66]:

1. For $k < m$ such that $r_k + k > m$

$$\mathcal{E}_0 \leq \left( 1 - \frac{k}{m} \right)^{-1} \left( 1 - \frac{m}{k + r_k} \right)^{-1}.$$

2. For $k \geq m$ it holds that

$$\frac{1}{m} \sum_i \mathcal{L}_{i,\delta} \geq \frac{m}{k} \left( \frac{k-m}{k - m + r_k} \right).$$

Note that in this case $c_1' m^{\frac{1}{d-1}} \leq \sqrt{T} = \sqrt{\frac{2}{\tau_m^2}} \leq c_2' m^{\frac{1}{d-1}}$, where $c_2' = \sqrt{\frac{2}{C_1^2}}$ (and similarly for indices 1 and 2 swapped with inequalities reversed).
Theorem 28 shows that for $i$ is such that $i \leq (k\sqrt{T})$, we have that for $\tilde{\lambda}_i$ it holds that

$$\frac{\tilde{\lambda}_i}{\tilde{\lambda}_1} \geq \left( \frac{1}{1 + \frac{k}{\sqrt{T}}} \right)^{k\sqrt{T}} \geq \exp(-k^2).$$

Take $k = (1+\varepsilon)m$. We want to show that there is a constant $B > 0$ that depends on the dimension $d$ and the constant $C$ such that

$$r_k \leq Bm.$$

Let $\tilde{k} = ((1+\varepsilon)m)^{\frac{1}{d-1}}$. Note that by an analogous proof to Theorem 28, we have that for the size of eigenvalues it holds

$$\frac{\tilde{\lambda}_{\tilde{k}+j}}{\tilde{\lambda}_{\tilde{k}}} \leq \left( \frac{T}{T + (\tilde{k} + j - 1 + \frac{d}{2} - \frac{1}{2})} \right)^{\frac{j-1}{2}}.$$

For $l\tilde{k} \le j \le (l+1)\tilde{k}$, we have that

$$\frac{\tilde{\lambda}_{\tilde{k}+j}}{\tilde{\lambda}_{\tilde{k}}} \le \left(\frac{T}{T + (\tilde{k}+j-1+\frac{d}{2}-\frac{1}{2})}\right)^{\frac{j-1}{2}} < \left(\frac{T}{T+((l+1)\tilde{k})}\right)^{\frac{l\tilde{k}-1}{2}} < \left(\frac{T}{T+((l+1)\tilde{k})}\right)^{\frac{(l+1)\tilde{k}}{4}}.$$

We have that $\tilde{k} = \frac{1}{c'}(1+\varepsilon)^{\frac{1}{d-1}} c' m^{\frac{1}{d-1}} = \frac{1}{c'}(1+\varepsilon)^{\frac{1}{d-1}}\sqrt{T} = \alpha\sqrt{T}$ for $\alpha = \frac{1}{c'}(1+\varepsilon)^{\frac{1}{d-1}}$. Therefore, we have that

$$\frac{\tilde{\lambda}_{\tilde{k}+j}}{\tilde{\lambda}_{\tilde{k}}} < \left(\frac{T}{T+((l+1)\tilde{k})}\right)^{\frac{(l+1)\tilde{k}}{4}} = \left(\frac{T}{T+((l+1)\alpha\sqrt{T})}\right)^{\frac{(l+1)\alpha\sqrt{T}}{4}} \to \exp\left(-\frac{(l+1)^2\alpha^2}{4}\right),$$

as $m \to \infty$ i.e. $\sqrt{T} \to \infty$. Therefore, for $m$ large enough, for all $l$ we have that if $l\tilde{k} \le j \le (l+1)\tilde{k}$ then

$$\frac{\tilde{\lambda}_{\tilde{k}+j}}{\tilde{\lambda}_{\tilde{k}}} \le \frac{1}{(l+1)^{d+2}}.$$

The last inequality is true because we can take $m$ large enough so that $\alpha\sqrt{T}$ and $\frac{\sqrt{T}}{\alpha}$ are both greater than $400d$. Then we have that for all $l$

$$\frac{\tilde{\lambda}_{\tilde{k}+j}}{\tilde{\lambda}_{\tilde{k}}} < \left(\frac{T}{T+((l+1)\alpha\sqrt{T})}\right)^{\frac{(l+1)\alpha\sqrt{T}}{4}} < \left(\frac{1}{1+\frac{l+1}{100d}}\right)^{(l+1)100d} < \frac{1}{(l+1)^{d+2}}.$$

The last inequality holds because of the following

$$\left(1+\frac{l+1}{100d}\right)^{(l+1)100d} > \left(\frac{l+1}{100d}\right)^{d+2}\binom{(l+1)100d}{d+2} > (l+1)^{d+2}\frac{1}{(100d)^{d+2}}(100d)^{d+2} > (l+1)^{d+2}.$$

Therefore, we can bound $r_k$ as follows

$$r_k = \frac{\sum_{i>k}\lambda_k}{\lambda_k}$$

$$= \tilde{N}(d,\tilde{k}) + \sum_{j=1}^{\infty} N(d,\tilde{k}+j)\frac{\tilde{\lambda}_{\tilde{k}+j}}{\tilde{\lambda}_{\tilde{k}}}$$

$$< N(d,\tilde{k}) + \sum_{j=1}^{\infty} N(d,\tilde{k}+j)\frac{\tilde{\lambda}_{\tilde{k}+j}}{\tilde{\lambda}_{\tilde{k}}} < (\tilde{k})^{d-2} + \sum_{l=1}^{\infty}((l+1)\tilde{k})^{d-2}(\tilde{k})\frac{1}{(l+1)^{d+2}}$$

$$< \tilde{k}^{d-1}\left(\sum_{l=1}^{\infty}\frac{1}{(l+1)^4}\right) < \tilde{k}^{d-1}C = C(1+\varepsilon)m.$$

Therefore, applying claim 1, we have that $k - m + r_k < (C(1+\varepsilon)+\varepsilon)m$, so we have that $\frac{k-m}{k-m+r_k} > \frac{\varepsilon}{(C(1+\varepsilon)+\varepsilon)}$, so it follows that $\mathcal{E}_0 > (1-\frac{(1+\varepsilon)\varepsilon}{(C(1+\varepsilon)+\varepsilon)})^{-1} > 1$.

Note that the above proof actually shows that for any $k_1 \le k$, we have that $r_{k_1} \le Cm$. Now we want to show that $r_k$ is lower bounded and that we can take $k < m$ such that $k + r_k > m$. Again apply Theorem 28, we can generalize

$$\frac{\tilde{\lambda}_i}{\tilde{\lambda}_1} \ge \left(\frac{1}{1+\frac{k}{\sqrt{T}}}\right)^{k\sqrt{T}} \ge \exp(-k^2).$$

to other ratios as well. Let $j \le k\sqrt{T}$ and let $k\sqrt{T} \le i \le (k+1)\sqrt{T}$. Then we have that

$$\frac{\tilde{\lambda}_i}{\tilde{\lambda}_j} \ge \left(\frac{1}{1+\frac{2k}{\sqrt{T}}}\right)^{(k+1)\sqrt{T}} \ge \exp(-2k(k+1)).$$

Note that $c_1' m^{\frac{1}{d-1}} \le \sqrt{T} = \sqrt{\frac{2}{\tau_m^2}} \le c_2' m^{\frac{1}{d-1}}$, where $c_2' = \sqrt{\frac{2}{C_1^2}}$ (and similarly for indices 1 and 2 swapped with inequalities reversed). Therefore, we have that the following bound holds on $r_k$ for any $k < m$:

$$r_k = \frac{\sum_{i>k} \lambda_i}{\lambda_{k+1}} = \tilde{N}(d, (k\tilde{+}1)) + \sum_{j=1}^{\infty} N(d, (k\tilde{+}1)+j) \frac{\tilde{\lambda}_{(k\tilde{+}1)+j}}{\tilde{\lambda}_{(k\tilde{+}1)}}.$$

Note that since $\sqrt{T} \ge c_1' m^{\frac{1}{d-1}}$ then $N_{\sqrt{T}} \ge n_l(c_1')^{d-1} m$, so $\tilde{m} \le \frac{1}{c_1'} \frac{1}{n_l^{\frac{1}{d-1}}}$. Therefore, if we take $j = \tilde{k}$, $l = \frac{1}{c_1'} \frac{1}{n_l^{\frac{1}{d-1}}}$, and any $l\sqrt{T} \le i \le (l+1)\sqrt{T}$, we have that (since $\exp(-2l(l+1)) < \exp(-l^2)$)

$$\frac{\tilde{\lambda}_i}{\tilde{\lambda}_j} \ge \exp(-2l(l+1)).$$

Note that there is at least $N_{(l+1)\sqrt{T}} - N_{l\sqrt{T}} \ge n_l \left((l+1)^{d-1} - l^{d-1}\right) m$ such indices $i$, we have that

$$r_k = \frac{\sum_{i>k} \lambda_i}{\lambda_{k+1}} = \tilde{N}(d, (k\tilde{+}1)) + \sum_{j=1}^{\infty} N(d, (k\tilde{+}1)+j) \frac{\tilde{\lambda}_{(k\tilde{+}1)+j}}{\tilde{\lambda}_{(k\tilde{+}1)}}.$$
$$> \sum_{l\sqrt{T} \le i \le (l+1)\sqrt{T}} N(d, i) \frac{\tilde{\lambda}_i}{\tilde{\lambda}_k}$$
$$> n_l \left((l+1)^{d-1} - l^{d-1}\right) m \exp(-2l(l+1)) > \beta m,$$

where $\beta = n_l \left((l+1)^{d-1} - l^{d-1}\right) \exp(-2l(l+1))$. Note that $\beta$ is independent of $k$, so in particular for $k = (1 - \beta/2)m$, we will have $r_k + k > m$. Therefore we can use the upper bound on $\mathcal{E}_0$ to see that

$$\mathcal{E}_0 \le \frac{2^2}{\beta^2}(1 + \beta/2).$$

Since $\beta$ only depends on $d$ and $C_1$ and $C_2$, this finishes the proof. $\qquad \square$

In particular, Lemma 18 shows that when $\tau_m = o(m^{-\frac{1}{d-1}})$ or $\tau_m = \omega(m^{-\frac{1}{d-1}})$, then $\limsup \tilde{\mathbb{R}}(\hat{f}_0) \le \infty$.

**Lower bound for the test risk of Gaussian KRR**  Now we will show lower bounds for the test risk of Gaussian KRR.

**Lemma 19** (Lower bound for the test risk). Under Assumption 2, the following bounds hold for the risk $\tilde{R}(\hat{f}_0)$ of the minimum norm interpolating solution of Gaussian KRR:

1. If $\tau_m = o(m^{-\frac{1}{d-1}})$, then $\tilde{R}(0) \le \liminf \tilde{R}(\hat{f}_0) \le \limsup_{m\to\infty} \tilde{R}(\hat{f}_0) < \infty$. More precisely, if $\tau_m \le m^{-\frac{1}{d-1}} t(m)$, where $t(m) \to 0$ as $m \to \infty$, then there is a scalar $c_d$ that depends only on the dimension so that for all such $t(m)$ there is $m_0$ that depends on $t(m)$ such that for all $m > m_0$ we have $\tilde{R}(\hat{f}_0) > \sigma^2 + (1 - c_d t(m)^{\frac{d-1}{2}})\|f^*\|^2$.

2. If $\tau_m = \omega(m^{-\frac{1}{d-1}})$, then $\lim_{m\to\infty} \tilde{R}(\hat{f}_0) = \infty$. More precisely, if $\tau_m \ge m^{-\frac{1}{d-1}} t(m)$, where $t(m) \to \infty$ as $m \to \infty$ then there is an integer $m_0$ that depends on the dimension and $t(m)$ such that for $m > m_0$, we have that $t R(\hat{f}_0) \to \sqrt{m}(\sigma^2)$ (note that $\tilde{R}(0)$ is bounded). Hence, for large enough $m$ we have $\tilde{R}(\hat{f}_0) > \tilde{R}(0)$.

3. If $\tau_m = \Theta(m^{-\frac{1}{d-1}})$, then $\limsup_{m \to \infty} R(\hat{f}_0) < \infty$. Moreover, suppose that $C_1 m^{-\frac{1}{d-1}} \le \tau_m \le C_2 m^{-\frac{1}{d-1}}$ for some constants $C_1$ and $C_2$, then there exist $\eta, \mu > 0$ that depend only on $d, C_1$, and $C_2$, such that for all $m$ we have $\tilde{R}(\hat{f}_0) > \mu \|f^*\|^2 + (1+\eta)\sigma^2$. Consequently, $\tilde{R}(\hat{f}_0) > \tilde{R}(0)$ as long as $\sigma^2 > \frac{1-\mu}{\eta}\|f^*\|^2$.

*Proof.* All three proofs will similarly follow by directly analyzing Equation (2).

**Part 1:** Note that for any $i$, we have that $\tilde{\lambda}_i + \kappa_0 \le \tilde{\lambda}_1 + \kappa_0$, so we have that $\mathcal{L}_i \ge \frac{\tilde{\lambda}_i}{\tilde{\lambda}_1}\mathcal{L}_1$. From Theorem 28 we know that for $j \le \sqrt{T}t(m)^{\frac{1}{2}}$, we have that

$$\frac{\tilde{\lambda}_j}{\tilde{\lambda}_1} \ge \exp(-t(m)),$$

so there is $m_0$ that depends on $t(m)$ such that for all $m > m_0$

$$\tilde{\lambda}_j \ge \frac{1}{2}\tilde{\lambda}_1,$$

for all $j \le \sqrt{T}t(m)^{\frac{1}{2}}$. Therefore we have that

$$m = \sum_i N(d,i)\mathcal{L}_i > \left(N(d,1) + \dots N(d,\sqrt{T}t(m))\right)\frac{1}{2}\mathcal{L}_1.$$

Note that $\left(N(d,1) + \dots N(d,\sqrt{T}t(m))\right) > c_l \left(\sqrt{T}t(m)^{\frac{1}{2}}\right)^{d-1} = c_l' m(t(m))^{-\frac{d-1}{2}}$, where $c_l' = \sqrt{2}c_l$. Therefore, we have that $\mathcal{L}_1 < \frac{2}{c_l'}(t(m))^{\frac{d-1}{2}}$. Therefore, we have that

$$(1 - \mathcal{L}_i)^2 \ge (1 - \mathcal{L}_1)^2 > \left(1 - \frac{2}{c_l'}(t(m))^{\frac{d-1}{2}}\right).$$

From Equation (2), we have that then

$$\tilde{R}(\hat{f}_0) \ge \mathcal{E}_0\sigma^2 + \mathcal{E}_0(1 - c_d t(m)^{\frac{d-1}{2}})\|\beta\|^2.$$

This shows the first claim.

For the second part, note the following property of $\mathcal{L}_{i,\delta} = \frac{\lambda_i}{\lambda_i + \kappa_\delta}$: if for $j > i$, we have that $\mathcal{L}_{i,\delta} > \mathcal{L}_{j,\delta}$. Let $\frac{\lambda_j}{\lambda_i} > c$ for some fixed $c > 0$ and $j > i$, then we have that

$$\mathcal{L}_{j,\delta} > c\mathcal{L}_{i,\delta}.$$

By Theorem 28, we have that for $j \le \sqrt{T}g(m)$ with $g(m) \to 0$ as $m \to \infty$

$$\frac{\tilde{\lambda}_{\tilde{k}+j}}{\tilde{\lambda}_{\tilde{k}}} > \exp\left(-5g(m)\right).$$

So there is $m > m_0$ such that for $m > m_0$

$$\frac{\tilde{\lambda}_{\tilde{k}+j}}{\tilde{\lambda}_{\tilde{k}}} > \exp\left(-5g(m)\right) > \frac{1}{2}.$$

Note that

$$\frac{m}{m - \sum_{i=1}^{\infty} \mathcal{L}_{i,\delta}^2}(1 - \mathcal{L}_{j,\delta})^2 > 1 \text{ if}$$

$$\sum_{i=1}^{\infty} \mathcal{L}_{i,\delta}^2 > m\mathcal{L}_{j,\delta}(2 - \mathcal{L}_{j,\delta})$$

For this it suffices to have

$$\sum_{i=1}^{\infty} \mathcal{L}_{i,\delta}^2 > m\mathcal{L}_{1,\delta}\left(2 - \mathcal{L}_{1,\delta}\right),$$

since $\mathcal{L}_{i,\delta}$ are decreasing. Note that there is at least $N(d,1) + \cdots + N(d, \sqrt{T}g(m)) > n_l(\sqrt{T}g(m))^{d-1} > c_u m \left(\frac{g(m)}{t(m)}\right)^{d-1}$ of indices $i$ such that

$$\frac{\lambda_i}{\lambda_1} = \frac{\tilde{\lambda}_{1+j}}{\tilde{\lambda}_1} > \frac{1}{2}.$$

Therefore, if we select $g(m) = \sqrt{t(m)}$, then we have that

$$\sum_{i=1}^{\infty} \mathcal{L}_{i,\delta}^2 > \frac{1}{2}c_u m \left(\frac{1}{t(m)}\right)^{\frac{d-1}{2}} > m\mathcal{L}_{1,\delta}\left(2 - \mathcal{L}_{1,\delta}\right).$$

This implies that for all $j$ simultaneously

$$\frac{m}{m - \sum_{i=1}^{\infty} \mathcal{L}_{i,\delta}^2}(1 - \mathcal{L}_{j,\delta})^2 > 1.$$

This translates to

$$\tilde{R}(\hat{f}_0) > \tilde{R}(0) + \left(\frac{m}{m - \sum_{i=1}^{\infty} \mathcal{L}_{i,\delta}^2} - 1\right)\sigma^2.$$

This finishes the proof of the first part.

**Part 2:** Note that we showed that for $m$ large enough, $\mathcal{E}_0 > \sqrt{m}$. This shows that

$$\tilde{R}(\hat{f}_0) = \mathcal{E}_0\left(\sum_i (1 - \mathcal{L}_0)^2 v_i^2 + \sigma^2\right) > \sqrt{m}\sigma^2.$$

So, there is $m_0$ that depends only on $t(m)$ such that for $m > m_0$ we have that $\tilde{R}(\hat{f}_0) > \sqrt{m}\sigma^2$.

**Part 3:** Note first that $\mathcal{L}_i \geq \mathcal{L}_j$ for $i \leq j$. Let $T = \frac{2}{\tau_m^2}$. From Lemma 26 it follows that

$$n_l(\alpha\sqrt{T})^{d-1} \leq N(d,1) + \cdots + N(d, \alpha\sqrt{T}) \leq n_u(\alpha\sqrt{T})^{d-1}.$$

Take $\alpha$ such that $N(d,1) + \cdots + N(d, \alpha\sqrt{T}) > 2m$. Then, we have that

$$m = \sum_i N(d,i)\mathcal{L}_i > (N(d,1) + \cdots + N(d, \alpha\sqrt{T}))\mathcal{L}_{\alpha\sqrt{T}} > 2m\mathcal{L}_{\alpha\sqrt{T}}$$

$$\mathcal{L}_{\alpha\sqrt{T}} \leq \frac{1}{2} \implies \lambda_{\alpha\sqrt{T}} \leq \kappa_0.$$

Note that $(1 - \mathcal{L}_1)^2 < (1 - \mathcal{L}_i)^2$ and

$$(1 - \mathcal{L}_1)^2 = \frac{\kappa_0^2}{(\lambda_1 + \kappa_0)^2}.$$

If $\kappa_0 > \lambda_1$, then $2\kappa_0 > \lambda_1 + \kappa_0$, so $\frac{1}{2\kappa_0} < \frac{1}{\lambda_1 + \kappa_0}$. This shows that $\frac{\kappa_0^2}{(\lambda_1 + \kappa_0)^2} > \frac{1}{4}$. Otherwise $\kappa_0 \leq \lambda_1$, so $\kappa_0 + \lambda_1 \leq 2\lambda_1$. This implies that

$$\frac{\kappa_0^2}{(\kappa_0 + \lambda_1)^2} > \frac{\kappa_0^2}{4\lambda_1^2} > \frac{\lambda_{\alpha\sqrt{T}}}{4\lambda_1^2}.$$

From Theorem 28 it follows that $\frac{\lambda_{\alpha\sqrt{T}}}{4\lambda_1^2} > \exp(-\alpha^2)$. Since $\left(\frac{1}{n_u \frac{2}{C_1^2}}\right)^{\frac{1}{d-1}} < \alpha < \left(\frac{1}{n_l \frac{2}{C_2^2}}\right)^{\frac{1}{d-1}}$, we know that $\exp\left(-\alpha^2\right) > \exp\left(-\left(\frac{1}{n_l \frac{2}{C_2^2}}\right)^{\frac{2}{d-1}}\right)$, so $\mu = \exp\left(-\left(\frac{1}{n_l \frac{2}{C_2^2}}\right)^{\frac{2}{d-1}}\right)$. Combining Lemma 18 and Lemma 19, we get Theorem 3. $\qquad\square$

## C  The case of increasing dimension

In Appendix C.1 we will prove Theorem 7 and Theorem 9. In Appendix C.2, we will prove related claims about the eigenvalue multiplicities for relevant scalings of the dimension Corollary 12 and Corollary 13. In Appendix C.3, we will give a detailed proof of Corollary 13. In Appendix C.4, we will discuss when the conditions posed on kernels are satisfied.

### C.1  Proofs of Theorem 7 and Theorem 9

**Theorem 20** (Error scaling for any kernel ridge regression). Let $d$ and $m$ be any dimension and sample size. Define $L_m$, $U_m$, $k_m$, $N(i)$, $N_l$, and $\tilde{\lambda}_k$ as in Definition 5. Consider KRR with a kernel $K$ satisfying Assumption 6 for some $A$. Assume that for some integer $l$, the target function $f^*$ satisfies Assumption 4 with at most $N_l$ nonzero coefficients. Then, the risk of the minimum norm interpolating solution is bounded by the following:

$$\tilde{R}(\hat{f}_0) \leq \left(1 - \frac{L_m}{m}\right)^{-1} \left(1 - \frac{m}{U_m}\right)^{-1} \sigma^2 \tag{8}$$

$$+ B^2 \left(1 - \frac{L_m}{m}\right)^{-1} \left(1 - \frac{m}{U_m}\right)^{-1} \frac{A^2}{m^2} \left(\sum_{i=1}^l N(i) \frac{1}{\tilde{\lambda}_i^2}\right). \tag{9}$$

Additionally, we can get an alternative bound if we swap $\left(\sum_{i=1}^l N(d,i)\frac{1}{\tilde{\lambda}_i^2}\right)$ with $\left(\sum_{i=1}^l N(d,i)\frac{1}{\tilde{\lambda}_i}\right)$, i.e.

$$\tilde{R}(\hat{f}_0) \leq \left(1 - \frac{L_m}{m}\right)^{-1} \left(1 - \frac{m}{L_m}\right)^{-1} \sigma^2 \; + B^2 \left(1 - \frac{L_m}{m}\right)^{-1} \left(1 - \frac{m}{L_m}\right)^{-1} \frac{A}{m^2} \left(\sum_{i=1}^{L_m} N(d,i)\frac{1}{\tilde{\lambda}_i}\right).$$

**Proofs of Theorem 7 and Theorem 20:**  Note first that Equation (2) implies that

$$\tilde{R}(\hat{f}_0) = \sum_i \mathcal{E}_0 \left(1 - \mathcal{L}_{i,0}\right)^2 \beta_i^2 + \mathcal{E}_0 \sigma^2.$$

Let $\mathcal{L}_i = \frac{\tilde{\lambda}_i}{\tilde{\lambda}_i + \kappa_0}$. Then, we can rewrite it as

$$\tilde{R}(\hat{f}_0) = \sum_i \mathcal{E}_0 N(d,i) \left(1 - \mathcal{L}_i\right)^2 \beta_i^2 + \mathcal{E}_0 \sigma^2,$$

with a slight of abuse of notation for $\beta_i$. From Equation (5), we have that for $k < m$

$$\mathcal{E}_0 \leq \left(1 - \frac{k}{m}\right)^{-1} \left(1 - \frac{m}{k + r_k}\right)^{-1}.$$

Take $k = L_m$. Note that by Definition 5, $L_m < m$ Then $\lambda_{k+1} = \tilde{\lambda}_{k_m+1}$ and it repeats $N(d, k_m + 1)$ times. Therefore, we have that

$$r_k = N(d, k_m + 1) + \sum_{i=1}^{\infty} N(d, k_m + i + 1)\frac{\tilde{\lambda}_{k_m+i+1}}{\tilde{\lambda}_{k_m+1}}.$$

Therefore, $r_k + k > N(d, k_m + 1) + L_m = L_m > m$. Therefore, we can apply Equation (5). We get that

$$\mathcal{E}_0 \leq \left(1 - \frac{k}{m}\right)^{-1} \left(1 - \frac{m}{k + r_k}\right)^{-1} = \left(1 - \frac{L_m}{m}\right)^{-1} \left(1 - \frac{m}{L_m}\right)^{-1}.$$

Note now that

$$m = \sum_i N(d,i)\mathcal{L}_i.$$

Note that $\tilde{\lambda}_i + \kappa_0 > \kappa_0$ so we have that $\mathcal{L}_i < \frac{\tilde{\lambda}_i}{\kappa_0}$. Therefore, we have that

$$m = \sum_i N(d,i)\mathcal{L}_i < \frac{1}{\kappa_0}\left(\sum_i N(d,i)\tilde{\lambda}_i\right) < \frac{A}{\kappa_0}.$$

Therefore, we have that $\kappa_0 < \frac{A}{m}$. Since $(1-\mathcal{L}_i)^2 = \frac{\kappa_0^2}{(\kappa_0+\tilde{\lambda})^2} < \frac{\kappa_0^2}{\tilde{\lambda}^2}$. Since there are only $l_d$ nonzero $\beta_i$, we have that

$$\sum_i N(d,i)(1-\mathcal{L}_i)^2\beta_i^2 \le B^2\frac{A^2}{m^2}\left(\sum_{i=1}^l N(d,i)\frac{1}{\tilde{\lambda}_i^2}\right).$$

Going back to Equation (2), we have

$$\tilde{R}(\hat{f}_0) = \mathcal{E}_0\sum_i(1-\mathcal{L}_i)^2 N(d,i)\beta_i^2 + \mathcal{E}_0\sigma^2$$

$$\le \mathcal{E}_0 B^2\frac{A^2}{m^2}\left(\sum_{i=1}^l N(d,i)\frac{1}{\tilde{\lambda}_i^2}\right) + \mathcal{E}_0\sigma^2$$

$$\le \left(1-\frac{L_m}{m}\right)^{-1}\left(1-\frac{m}{L_m}\right)^{-1}\sigma^2$$

$$+ \left(1-\frac{L_m}{m}\right)^{-1}\left(1-\frac{m}{L_m}\right)^{-1}B^2\frac{A^2}{m^2}\left(\sum_{i=1}^l N(d,i)\frac{1}{\tilde{\lambda}_i^2}\right)$$

This shows the first bound. For the second bound, note that

$$\sum_{i=1}^\infty \frac{\kappa_0^2}{(\kappa_0+\lambda_i)^2}\beta_i^2 \le \sum_{i=l+1}^\infty \beta_i^2 + \sum_{i=1}^l \beta_i^2\frac{1}{\sum_i\lambda_i}\frac{1}{\lambda_i}\left(\frac{\sum_{j>l}\lambda_j}{m}\right)^2.$$

So if we take $\beta$ to have only $l$ nonzero terms, we get

$$\sum_{i=1}^l \frac{\kappa_0^2}{(\kappa_0+\lambda_i)^2}\beta_i^2 \le \sum_{i=1}^l \beta_i^2\frac{1}{\sum_i\lambda_i}\frac{1}{\lambda_i}\left(\frac{\sum_{j>l}\lambda_j}{m}\right)^2.$$

From this, we have that

$$\sum_{i=1}^l \frac{\kappa_0^2}{(\kappa_0+\lambda_i)^2}\beta_i^2 \le \frac{A}{\lambda_i}\frac{1}{m^2}.$$

Therefore, we have that

$$N(d,i)\frac{\kappa_0^2}{(\kappa_0+\tilde{\lambda}_i)^2}\beta_i^2 \le N(d,i)\beta_i^2\frac{A}{\lambda_i}\frac{1}{m^2}.$$

Therefore, we have the improved inequality from

$$\sum_{i=1}^l N(d,i)(1-\mathcal{L}_i)^2\beta_i^2 \le \sum_{i=1}^l \beta_i^2 N(d,i)\frac{A}{\lambda_i}\frac{1}{m^2}.$$

This gives the second bound on the test risk, as we can bound

$$\sum_{i=1}^l N(d,i)(1-\mathcal{L}_i)^2\beta_i^2 \le \frac{B^2 A}{m^2}\left(\sum_{i=1}^l N(d,i)\frac{1}{\lambda_i}\right).$$

[Lower bound for test risk in increasing dimension] Let $k_m$ and $L_m$ be as in Definition 5. Consider learning a target function $f^*$, with some sample size $m$. Let $K$ be a kernel satisfying Assumption 8 for some $A$ and $b$. Consider the minimum norm interpolating solution of KRR (with any data distribution) with kernel $K$. Then, for the risk of minimum norm interpolating solution, the following lower bound holds:

$$\tilde{R}(\hat{f}_0) > \left(1-\left(\frac{b}{b+1}\right)^2\frac{L_m}{m}\right)^{-1}\sigma^2.$$

**Proof of Theorem 9:** By Theorem 3 from Zhou et al. [66], if $k$ is the first $k < m$ such that $k + br_k \geq m$, then

$$\mathcal{E}_0 \geq \left(1 - \left(\frac{b}{b+1}\right)^2 \frac{k}{m}\right)^{-1}.$$

Therefore, it suffices to show that the first such $k$ is actually $L_m = N(d, 1) + \cdots + N(d, k_m)$. Note that

$$r_k = \frac{\sum_{i>k} \lambda_i}{\lambda_{k+1}} < \max_{i \leq k} \left(\frac{1}{\lambda_i}\right) \left(\sum_{i>k} \lambda_i\right) < A \max_{i \leq k} \left(\frac{1}{\lambda_i}\right).$$

Since $\max_{i \leq k_m} \left(\frac{1}{\lambda_i}\right) < \frac{m - L_m}{b}$, for $l < N(d, 1) + \cdots + N(d, k_m)$, we have that $br_l + l \leq N(d, 1) + \cdots + N(d, k_m) - 1 + b \max_{i \leq k_m} \left(\frac{1}{\lambda_i}\right) < L_m + m - L_m \leq m$, so the first $l$ for which $r_l + l > m$ is $l = L_m = N(d, 1) + \cdots + N(d, k_m)$. Plugging in $k = L_m$ we get that $\mathcal{E}_0 \geq \left(1 - \left(\frac{b}{b+1}\right) \frac{L_m}{m}\right)^{-1}$, so from Equation (6), we have $\tilde{R}(\hat{f}_0) \geq \left(1 - \left(\frac{b}{b+1}\right)^2 \frac{L_m}{m}\right)^{-1} \sigma^2$.

## C.2 Dot-product kernels on the sphere

First we will prove results about the eigenvalue multiplicites Corollary 12 and Corollary 13.

**Theorem 21** (Log-scaling multiplicity). Let $d = \log_2 m$, i.e. $m = 2^d$, and let $k_d$ be an index for which the following holds:

$$N(d, 1) + \cdots + N(d, k_m) < m$$
$$N(d, 1) + \cdots + N(d, k_m + 1) \geq m.$$

Then the following hold:

1. $\frac{dN(d, \frac{d}{5})}{2^d}$ is decreasing and has $\lim_{d \to \infty} \frac{dN(d, \frac{d}{5})}{2^d} \to 0$. This also holds for $\frac{dN(d, k)}{2^d}$ with $k = f(d) \leq \frac{d}{5}$ for all $d$.

2. $\frac{N(d, \frac{d}{2})}{2^d}$ is increasing and has $\lim_{d \to \infty} \frac{N(d, \frac{d}{2})}{2^d} \to \infty$. This also holds for $\frac{N(d, k)}{2^d}$ with $k = f(d) \geq \frac{d}{2}$ for all $d$.

3. There is an absolutet constant $d_0$ such that for $d > d_0$, we have that $\frac{d}{5} \leq k_m \leq \frac{d}{2}$.

4. There are absolute constants $c_{l-1} = \frac{1}{54}, c_l = \frac{1}{9}$, and $c_{l+1} = \frac{2}{3}$ such that $N(d, k_m + i) > c_{l+i} m$, for all $i \in \{\pm 1, 0\}$. Additionally, there are constants $c_{u-1} = \frac{1}{3}, c_u = 1, c_{u+1} = 6$, such that $N(d, k_m + i) < c_{u+i} m$ for all $i \in \{\pm 1, 0\}$.

Furthermore, if $\frac{d}{\log m} = \Theta(1)$, then we will also have $L_m = \Theta(m)$ and $L_m = \Theta(m)$.

*Proof.* We will split the proof into three parts. Note first that $N(d, k)$ increases as $k$ increases. This can be seen from the ratio of consecutive multiplicities

$$\frac{N(d, k+1)}{N(d, k)} = \frac{2k+d}{2k+d-2} \frac{k+d-2}{k+1}.$$

**Part 1:** We will use Stirling's approximation to estimate $N(d, k)$. It states that $n! \approx \left(\frac{n}{e}\right)^n \sqrt{2\pi n}$. Therefore, we have that

$$N(d, k) = \frac{(2k+d-2)(k+d-3)!}{k!(d-2)!} \approx \frac{\sqrt{2\pi(k+d-3)} \left(\frac{k+d-3}{e}\right)^{k+d-3} (2k+d-2)}{\left(\frac{k}{e}\right)^k \left(\frac{d-2}{e}\right)^{d-2} \sqrt{k}\sqrt{d-2}}.$$

Since $N(d, k)$ is increasing in $k$ and $k \le \frac{d}{5}$, so we can just plug in $k = \frac{d}{5}$ in the expression. Therefore

$$N(d, \frac{d}{5}) \approx \frac{\sqrt{2\pi(k+d-3)}\left(\frac{k+d-3}{e}\right)^{k+d-3}(2k+d-2)}{\left(\frac{k}{e}\right)^k \left(\frac{d-2}{e}\right)^{d-2}\sqrt{k}\sqrt{d-2}}$$

$$= \frac{\sqrt{2\pi(\frac{d}{5}+d-3)}\left(\frac{\frac{d}{5}+d-3}{e}\right)^{\frac{d}{5}+d-3}(2\frac{d}{5}+d-2)}{\left(\frac{\frac{d}{5}}{e}\right)^{\frac{d}{5}} \left(\frac{d-2}{e}\right)^{d-2}\sqrt{\frac{d}{5}}\sqrt{d-2}}$$

$$= C\frac{\sqrt{d}d d^{\frac{d}{5}+d-3}\left(\frac{\frac{6}{5}-\frac{3}{d}}{e}\right)^{\frac{6}{5}d-3}}{\sqrt{d}\left(\frac{d}{5e}\right)^{\frac{d}{5}}\left(\frac{d}{e}\right)^{d-2}\sqrt{d}\sqrt{d}}$$

$$= C\frac{5^{\frac{d}{5}}\left(\frac{6}{5}\right)^{\frac{6d}{5}}}{\sqrt{d}} = \frac{\left(5^{1/5}(\frac{6}{5})^{6/5}\right)^d}{\sqrt{d}}.$$

Note that $\left(5^{1/5}(\frac{6}{5})^{6/5}\right) < 1.8$, so $\frac{\left(5^{1/5}(\frac{6}{5})^{6/5}\right)^d}{\sqrt{d}} < \frac{1.8^d}{\sqrt{d}}$. In particular, we have that

$$\frac{dN(d, \frac{d}{5})}{2^d} < \frac{\sqrt{d}(1.8)^d}{2^d} \to 0.$$

**Part 2:** The same calculation in this case gives

$$N(d, \frac{d}{2}) \approx C\frac{\left(2^{\frac{1}{2}}(\frac{3}{2})^{\frac{3}{2}}\right)^d}{\sqrt{d}}$$

but note that $\left(2^{\frac{1}{2}}(\frac{3}{2})^{\frac{3}{2}}\right) > 2.5$, so $\frac{N(d, \frac{d}{2})}{2^d} > \left(\frac{2.5}{2}\right)^d \frac{1}{\sqrt{d}} \to \infty$.

**Part 3:** Note that if $k_m \le \frac{d}{5}$ then for $d$ large enough

$$N(d, 1) + \cdots + N(d, k_m) \le k_m N(d, k_m) \le dN(d, k_m) < d.$$

Similarly, note that if $k_m > \frac{d}{2}$, then $N(d, k_m) > d$ for $d$ large enough. Both of these imply hat $\frac{d}{5} \le k_m \le \frac{d}{2}$ when $d > d_0$ ($d_0$ is determined by when the inequalities above start holding).

**Part 4:** Note that when $\frac{d}{5} \le k \le \frac{d}{2}$, we have that $6 > \frac{N(d, k+1)}{N(d, k)} = \frac{2k+d}{2k+d-2}\frac{k+d-2}{k+1} > 3$ for $d$ large enough, since the firs ratio is close to 1 and the second is $1 + \frac{d-2}{k+1}$. This implies that $\frac{N(d, k)}{N(d, k+1)} < \frac{1}{3}$ whenever $d$ is large enough (larger than an absolute constant) and $k \le \frac{d}{2}$. Note now that

$$m < N(d, 1) + \cdots + N(d, k_m + 1) < N(d, k_m + 1)\left(1 + \frac{1}{3} + (\frac{1}{3})^2 + \ldots\right) < N(d, k_m + 1)\frac{3}{2}$$

$$\frac{2}{3}m < N(d, k_m + 1).$$

This now implies that $N(d, k_m) > \frac{1}{6}N(d, k_m + 1) > \frac{1}{9}m$, $N(d, k_m - 1) > \frac{1}{54}m$. Similarly we have that $N(d, k_m) < m$, so $N(d, k_m + 1) < 6m$ and $N(d, k_m - 1) < \frac{1}{3}m$.

Note that the same proof works even if $d$ is not exactly equal to $\log_2 m$. This shows that whenever $\frac{d}{\log m} = \Theta(1)$, we have that $L_m = \Theta(m)$ and $L_m = \Theta(m)$. $\qquad \square$

**Remark 22.** Note that this does not say that we will always have tempered overfitting for $\frac{d}{\log m} = \Theta(1)$ as sometimes it can happen that $L_m \to m$, so we cannot apply Equation (5) and Equation (6). In this case we expect the overfitting to be catastrophic.

**Theorem 23** (Sub-polynomial scaling multiplicity). Let $K$ be any dot product kernel on the sphere $\mathbb{S}^{d-1}$ (with nonzero eigenvalues). Let $l \in \mathbb{N}$ and let $m = 2^{2^{2l}}$ and let $d = 2^{2^l}$. This corresponds to the case $d = \exp\left(\sqrt{\log m}\right)$, which is sub-polynomial. Then, for the upper and lower index, $L_m$ and $L_m$, we have the following

$$\frac{L_m}{m} \leq \frac{3}{2\log m} \quad \text{and} \quad \frac{m}{L_m} \leq \frac{1}{d^{0.89}}.$$

Additionally, for $k = \alpha k_m$, where $\alpha$ is a constant, we have that $N(d, k) \approx m^\alpha$, where $\approx$ means up to sub-polynomial factors.

*Proof.* Let $k_m$ be the maximum $k$ for which

$$N(d, 1) + \cdots + N(d, k_m) < m$$
$$N(d, 1) + \cdots + N(d, k_m + 1) \geq m.$$

We want to show that $N(d, k_m) = o(m)$ and $N(d, k_m + 1) = \Omega(m)$. Then we can take $k = N(d, 1) + \cdots + N(d, k_m)$ and so $\frac{k}{m} \to 0$ and

$$r_k = N(d, k_m + 1) + \sum_{j=2} N(d, k_m + j) \frac{\tilde{\lambda}_{k_m+j}}{\tilde{\lambda}_{k_m+1}} = \Omega(m)$$

$$\frac{m}{R_k} \to 0.$$

Note that we just need to estimate $N(d, k_m)$ precisely. Let $k = 2^l + l$. We want to show that $N(d, k) > m$. Note that by Stirling's approximation $n! \approx \left(\frac{n}{e}\right)^n \sqrt{2\pi n}$

$$
\begin{aligned}
N(d, k) &\approx \frac{\sqrt{2\pi(k+d-3)}\left(\frac{k+d-3}{e}\right)^{k+d-3}(2k+d-2)}{\left(\frac{k}{e}\right)^k \left(\frac{d-2}{e}\right)^{d-2}\sqrt{2\pi k}\sqrt{2\pi(d-2)}} \\
&= \frac{e}{\sqrt{2\pi}} \frac{d^{k+d-2}\sqrt{d}\sqrt{2\pi(1+\frac{k}{d}-\frac{3}{d})}\left(1+\frac{k}{d}-\frac{3}{d}\right)^{k+d-3}\left(1+\frac{2k}{d}-\frac{2}{d}\right)}{k^k(d-2)^{d-2}\sqrt{k}\sqrt{d-2}} \\
&= \frac{e}{\sqrt{2\pi}} \frac{d^{k+d-2}\sqrt{d}\sqrt{2\pi(1+\frac{2^l+l}{2^{2^l}}-\frac{3}{2^{2^l}})}\left(1+\frac{2^l+l}{2^{2^l}}-\frac{3}{2^{2^l}}\right)^{k+d-3}\left(1+\frac{2k}{d}-\frac{2}{d}\right)}{k^k d^{d-2}(1-\frac{2}{d})^{d-2}\sqrt{k}\sqrt{d}\sqrt{1-\frac{2}{d}}} \\
&= \frac{e}{\sqrt{2\pi}} \frac{d^k\sqrt{2\pi(1+\frac{k}{d}-\frac{3}{d})}\left(1+\frac{k}{d}-\frac{3}{d}\right)^{k+d-3}\left(1+\frac{2k}{d}-\frac{2}{d}\right)}{(1-\frac{2}{d})^{d-2}\sqrt{1-\frac{2}{d}}} \\
&= \frac{e}{\sqrt{2\pi}} \frac{\sqrt{2\pi(1+\frac{k}{d}-\frac{3}{d})}\left(1+\frac{2k}{d}-\frac{2}{d}\right)}{(1-\frac{2}{d})^{d-2}\sqrt{1-\frac{2}{d}}} \frac{d^k}{k^k\sqrt{k}}\left(1+\frac{k}{d}-\frac{3}{d}\right)^{k+d-3}.
\end{aligned}
$$

Only the term $\frac{d^k}{k^k\sqrt{k}}\left(1+\frac{k}{d}-\frac{3}{d}\right)^{k+d-3}$ is asymptotic here, as the rest tends to a constant as $d \to \infty$. Note that since for all $l > 5$ (i.e. $d$)

$$2 > \left(1+\frac{k}{d}-\frac{3}{d}\right)^{\frac{d}{k}} > 1.95,$$

we have that

$$\left(1+\frac{k}{d}-\frac{3}{d}\right)^{k+d-3} > (1.95)^{\frac{k^2}{d}+k-3\frac{k}{d}} > (1.95)^k > 2^{(0.9k)}$$

$$\left(1+\frac{k}{d}-\frac{3}{d}\right)^{k+d-3} < 2^{\frac{k^2}{d}+k-3\frac{k}{d}}.$$

Note that we have that then

$$\frac{d^k}{k^k\sqrt{k}}\left(1+\frac{k}{d}-\frac{3}{d}\right)^{k+d-3} < \frac{d^k}{k^k\sqrt{k}}2^{\frac{k^2}{d}+k-3\frac{k}{d}} = 2^{k\log d - k\log k - \frac{1}{2}\log k + \frac{k^2}{d}+k-\frac{3k}{d}}.$$

Consider now the expression $k\log d - k\log k - \frac{1}{2}\log k + \frac{k^2}{d}+k-\frac{3k}{d}$. For $k = 2^l + l - 1$, we have that

$$k\log d - k\log k - \frac{1}{2}\log k + \frac{k^2}{d}+k-\frac{3k}{d} = k\log d - k\log k + k - \frac{1}{2}\log k - \frac{3k}{d} + \frac{k^2}{d}$$

$$= \left(2^l + l - 1\right)2^l - \left(2^l + (l-1)\right)\left(l + \frac{l-1}{2^l} + o(\frac{1}{2^{2l}})\right) - \frac{1}{2}(l) + o(\frac{1}{2^l}) + 2^l + l - 1 + o(\frac{1}{2^l})$$

$$= 2^{2l} + 2^l(l-1) - 2^l(l) + 2^l - l + 1 - \frac{1}{2}l + l - 1 + o(1) = 2^{2l} - \frac{1}{2}l = \log_2 m - \log_2\log_2 m.$$

In particular, this implies that for $k \le 2^l + (l-1)$, we have that

$$N(d,k) < m2^{-\frac{1}{2}l} = o(m).$$

Now repeating the same computation for $k = 2^l + l$, we have that

$$\frac{d^k}{k^k\sqrt{k}}\left(1+\frac{k}{d}-\frac{3}{d}\right)^{k+d-3} > \frac{d^k}{k^k\sqrt{k}}2^{0.9k} = 2^{k\log d - k\log k - \frac{1}{2}\log k + 0.9k}.$$

$$k\log d - k\log k - \frac{1}{2}\log k + \frac{k^2}{d}+0.9k-\frac{3k}{d} = k\log d - k\log k + k - \frac{1}{2}\log k - \frac{3k}{d} + \frac{k^2}{d}$$

$$= \left(2^l + l\right)2^l - \left(2^l + l\right)\left(l + \frac{l}{2^l} + o(\frac{1}{2^{2l}})\right) - \frac{1}{2}(l) + o(\frac{1}{2^l}) + 0.9\cdot 2^l + 0.9l + o(\frac{1}{2^l})$$

$$= 2^{2l} + 2^l(l) - 2^l(l) + 0.9\cdot 2^l - l - \frac{1}{2}l + l + o(1) = 2^{2l} + 0.9\cdot 2^l - \frac{1}{2}l$$

$$= \log_2 m + 0.9\cdot\sqrt{\log_2 m} - \log_2\log_2 m.$$

This shows that for $k \ge 2^l + l$,

$$N(d,k) > m2^{0.9\cdot 2^l - \frac{1}{2}l} = \Omega(m).$$

To imply now that $k_m = k = 2^l + l - 1$, it suffices to show that $N(d,1) + \cdots + N(d,k) = o(m)$. But note that we have $k \ll d$, so

$$N(d,k-1) = \frac{2k+d-2}{2k+d}\frac{k+1}{k+d-2}N(d,k) < \frac{1}{d}N(d,k)$$

$$N(d,1) + \cdots + N(d,k-1) < \frac{k-1}{d}N(d,k)$$

$$N(d,1) + \cdots + N(d,k) = o(m).$$

Therefore, we have computed with proof that $k_m = 2^l + l - 1$. So we can now compute $L_m$ and $L_m$ with $L_m = N(d,1) + \cdots + N(d,k_m)$. Note that

$$\frac{N(d,k+1)}{N(d,k)} = \frac{2k+d}{2k+d-2}\frac{k+d-2}{k+1}.$$

Since $k \ll d$, note that $\frac{N(d,k+1)}{N(d,k)} > 2$, so we have that $\frac{N(d,k)}{N(d,k+1)} < \frac{1}{2}$, which implies that

$$N(d,1) + N(d,2) + \cdots + N(d,k_m) < \frac{3}{2}N(d,k_m) = o(m).$$

Therefore, we have that $\frac{L_m}{m} = \frac{N(d,1)+N(d,2)+\cdots+N(d,k_m)}{m} < \frac{3}{2}\frac{N(d,k_m)}{m} \le \frac{3}{2\log m}$. From what we computed previously, we have that

$$L_m > N(d,k) > m2^{0.9\cdot 2^l - \frac{1}{2}l}$$

$$\frac{m}{L_m} < \frac{1}{d^{0.9 - \frac{1}{2}\log\log d\log d}} < \frac{1}{d^{0.89}}.$$

Note finally that for $k = \alpha 2^l$, the leading term in the exponent of $N(d,k)$ as derived above is $\alpha 2^{2l}$, all other terms are at most $l2^l$, which is sub-polynomial in $m$, so this shows that $N(d,\alpha k) \approx m^\alpha$. $\qquad\square$

**Remark 24.** Note that for the polynomially increasing $d$, there are sequences $(m, d)$ for which we do not get benign overfitting, i.e. when $d = m^k$, for integer $k$. Similarly in this case, there are sequences of $(m, d)$ for which we do not get benign overfitting, but is much harder to identify them.

## C.3  Proof of Corollary 13

**Corollary 3** (Benign overfitting with Gaussian kernel and sub-polynomial dimension)**.** Let $K$ be the Gaussian kernel on the sphere $\mathbb{S}^{d-1}$ with a fixed bandwidth, and suppose that for $l \in \mathbb{N}$ the dimension and sample size scale as $d = 2^{2^l}$, $m = 2^{2^{2l}}$, i.e. $d = \exp\left(\sqrt{\log m}\right)$. Consider learning a sequence of target functions $f_d^*$ as in Assumption 4 with $S_d \leq m^{\frac{1}{4}}$. Then, we have that the minimum norm interpolating solution achieves the Bayes error in the limit $(m, d) \to \infty$. In particular, for $d \geq 4$ and $m \geq 16$ we have

$$\tilde{R}(\hat{f}_0) \leq \left(1 - \frac{1}{\log m}\right)^{-1} \left(1 - \exp\left(-0.89\sqrt{\log m}\right)\right)^{-1} \sigma^2 + 2B^2 \frac{1}{m}.$$

If we take $S_d = \text{poly}(d)$, then we can improve the $\frac{1}{m}$ error dependence to $\frac{1}{m^{2-\varepsilon}}$. Furthermore, we can improve the bound on $S_d$ by reducing the rate of convergence. Let $s_d$ be an integer such that $S_d \leq N_{s_d}$. Then error dependence is $\frac{1}{m^2} s_d N(d, s_d) d^{(1+\varepsilon)s_d} < \frac{1}{m^2} d^{2(1+\varepsilon)s_d}$.

**Proof:**  We have that from Theorem 23 that in this case $k_m = 2^l + l - 1 = \log d + \log \log d - 1$, $L_m = \Theta(\frac{m}{\log m})$ and $L_m \geq \Theta(md^{0.89})$. Therefore, we have that

$$\mathcal{E}_0 \leq \left(1 - \frac{1}{\log m}\right)^{-1} \left(1 - \exp\left(-0.89\sqrt{\log m}\right)\right)^{-1}.$$

To finish the proof, we need to estimate $N(d, 1)\frac{1}{\tilde{\lambda}_1^2} + \dots N(d, k_m)\frac{1}{\tilde{\lambda}_{k_m}^2}$. Note that the eigenvalues are sorted and $N(d, k-1) = o(N(d, k+1))$ since $k_m = o(d)$. This implies that it suffices to estimate $N(d, k_m)\frac{1}{\tilde{\lambda}_{k_m}^2}$. Note that for the first eigenvalue, we have from Corollary 31 for $\alpha = 1 + \frac{2}{\tau_m^2}$.

$$\frac{1}{d^\alpha} < \tilde{\lambda}_1$$

So, for $\tilde{\lambda}_k$, it holds that

$$\frac{\tilde{\lambda}_k}{\tilde{\lambda}_1} > \prod_{i=1}^{k} \frac{\frac{2}{\tau_m^2}}{\frac{2}{\tau_m^2} + 2(i + \frac{d}{2} - 1)}.$$

Therefore, we have that

$$\frac{1}{\tilde{\lambda}_k} < \frac{1}{\tilde{\lambda}_1} \prod_{i=1}^{k} \frac{\frac{2}{\tau_m^2} + 2(i + \frac{d}{2} - 1)}{\frac{2}{\tau_m^2}} = d^\alpha \left(\frac{\tau_m^2}{2}\right)^k \frac{\Gamma(d + 2k_m + \frac{2}{\tau_m^2})}{\Gamma(\frac{d}{2})\Gamma(d + \frac{2}{\tau_m^2})} < d^\alpha \left(\tau_m^2\right)^k \left(\frac{d}{2} + k + \frac{2}{\tau_m^2}\right)^k.$$

Note also that if $k = \beta k_m$ for some constant $\beta$ then

$$N(d, k) \approx m^\beta,$$

in the sense of Theorem 23. So, $N(d, k_m)\frac{1}{\tilde{\lambda}_{k_m}^2} < m^\beta d^{2k(1+\varepsilon)-2\alpha}$. We want to take $k$ as large as possible, so we will take it $k = \beta k_m$. Then $d^{2k} \approx m^{2\beta}$. Therefore, as long as $3\beta < 2$, we will have a polynomially scaling error. So we can take $\beta = \frac{2}{3}$. Note that with Theorem 20, we actually get even better dependence of the error, like $m^\beta d^{k(1+\varepsilon)-\alpha}$. Plugging the first estimate into Theorem 7, for $3\alpha = 1$, we have that

$$\tilde{R}(\hat{f}_0) \leq \left(1 - \frac{1}{\log m}\right)^{-1} \left(1 - \frac{1}{d^{0.89}}\right)^{-1} \sigma^2$$

$$+ B^2 \left(1 - \frac{1}{\log m}\right)^{-1} \left(1 - \frac{1}{d^{0.89}}\right)^{-1} \frac{1}{m^2} m^{3\beta} \quad \leq \left(1 - \frac{1}{\log m}\right)^{-1} \left(1 - \frac{1}{d^{0.89}}\right)^{-1} \sigma^2$$

$$+ B^2 \left(1 - \frac{1}{\log m}\right)^{-1} \left(1 - \frac{1}{d^{0.89}}\right)^{-1} \frac{1}{m}.$$

If we want to achieve $S_d = \text{poly}(d)$, note that we just need to take $k$ to not scale wit $d$. That is, if we want $\deg S_d = n$, then we can take $k = n$. The eigenvalue will be

$$\frac{1}{\tilde{\lambda}_n} < d^{\alpha+2n}.$$

The multiplicity will be $N(d, n) = \Theta(d^n)$. Note that then error scales as $\frac{d^{5n}}{m^2}$, i.e. since $d$ is sub poly $m$, we can take any such $n$.

## C.4 Kernel conditions

We have three main assumptions on kernels. First, the sum of its eigenvalues is bounded.

**Assumption 6** (Bounded sum of eigenvalues)**.** Assume that the kernel $K$ has a bounded sum of eigenvalues, i.e. there is a constant $A$ such that $\sum_{i=1}^{\infty} N(i)\tilde{\lambda}_i \leq A$. For a sequence of kernels $K^{(d)}$, assume that all such $A^{(d)}$ are bounded by some constant $A$.[2]

Second, there is a lower bound on the eigenvalues.

**Assumption 8** (Lower bound on eigenvalues)**.** Assumme that the kernel $K$ has eigenvalues that are not too small, i.e. there is a constant $b$ such that $\max_{i \leq k_m} \left(\frac{1}{\tilde{\lambda}_i}\right) < \frac{m - L_m}{b}$. For a sequence of kernels $K^{(d)}$, assume that for the corresponding $m = m(d)$ (since $d = d(m)$, we can also "invert" the dependence) all such $b^{(d)}$ are bounded below by some $b$.

And third, the eigenvalues don't decrease too fast.

**Assumption 10** (Eigenvalue decay)**.** The eigenvalues don't decrease too quickly, i.e. for $k_m$ as in Definition 5, we have that there is a constant $c$ such that $\max_{i \leq k_m} \left(\frac{1}{\tilde{\lambda}_i}\right) \leq cN(k_m)$. For increasing dimension, we require that $\max_{i \leq k_m} \left(\frac{1}{\tilde{\lambda}_i}\right) \leq cN(d, k_m)$.

Note that Assumption 6 will hold quite broadly, in particular at least for dot product kernels on the sphere (that are effectively the same kernel just in increasing dimensions). Note also that the Assumption 10 is almost always stronger than Assumption 8. This follows from the definition of $L_m$ and $k_m$, i.e. since $L_m = N(1) + \cdots + N(k_m)$, we have that $N(k_m) < m$ and as long as $L_m < \Theta(m)$, we can take $b$ large. This is usually the case (we show previously in Appendix C.2 for dot-product kernels on the sphere). For $d = \omega(\log m)$, we even have $L_m = o(m)$. So in particular $N(d, k_m) < \Theta(m - L_m)$ and for $d = \omega(\log m)$, we have $N(d, k_m) = o(m - L_m)$, so indeed Assumption 10 is stronger than Assumption 8.

Assumption 6 and Assumption 10 are also taken to be true in the literature on the polynomially scaling dimension [4, 64].

**First assumption:** For dot-product kernels on the sphere, we know that the eigenfunction $\phi_i$ are actually the spherical harmonics $Y_{ks}$. Let $K(x, y) = h(\|x - y\|)$. Note that for $x \in \mathbb{S}^{d-1}$, it holds that

$$\sum_{s=1}^{N(d,k)} Y_{ks}(x)^2 = N(d, k).$$

Therefore, since

$$K(x, y) = \sum_i \lambda_i \phi_i$$

$$h(0) = K(x, x) = \sum_k \sum_{s=1}^{N(d,k)} \tilde{\lambda}_k Y_{ks}(x)^2 = \sum_k N(d, k)\tilde{\lambda}_k.$$

So the assumption in Theorem 7 holds with $A = h(0)$. For the Gaussian kernel, this is $A = 1$. Similarly, $A = 1$ for Laplace kernel.

---

[2]Note that this assumption implicitly sets the scale of the kernel.

**Second Assumption:** For the eigenvalue assumption, $\max_{i \le k_m}\left(\frac{1}{\tilde{\lambda}_i}\right) < \frac{m-L_m}{Ab}$, note the following. We will usually have $L_m = o(m)$. In particular, whenever $d = \omega(\log m)$ for a dot-product kernel on the sphere this will hold.

The eigegnvalue assumption will hold for the Gaussian kernel in the non-integer polynomial regime, $d^\alpha = m$. In Appendix C.3, we showed that $\frac{1}{\tilde{\lambda}_k} < d^k$, so since $k_m = \lfloor \alpha \rfloor$, we will have $\frac{1}{\tilde{\lambda}_k} < d^\alpha - \Theta(d^{\lfloor \alpha \rfloor})$. Similarly, we can show this for the Gaussian kernel for $d = \log m$ and $d = \exp(\sqrt{\log m})$ regime. Note that in Corollary 31, we showed that $\tilde{\lambda}_1 > \frac{1}{d^\alpha}$ and $\tilde{\lambda}_k > \frac{1}{d^{\alpha+k}}$, so then $\frac{1}{\tilde{\lambda}_k}$ Additionally, in the $d = \exp(\sqrt{\log m})$ regime, we showed that $L_m = o(m)$ and that $\frac{1}{\tilde{\lambda}_k} = \Theta(L_m)$ for any $k = \Theta(k_m) < k_m$. The same proof shows that this assumption holds for any sub-polynomially scaling dimension and Gaussian kernel.

Furthermore, this assumption can be weakened to the following: there exists $b > 0$ such for all $k \le L_m$, $b\sum_{i>k} \lambda_i < \lambda_{k+1}(m-k)$. Additionally, since the assumption is specific to our approach, it seems to be possible to weaken it even further.

**Third assumption:** This assumption holds for dot-product kernels on the sphere. It was shown in Zhang et al. [64] (Lemma 5.2.1) under another assumption similar to (1) that it holds for polynomially scaling dimension. It is also assumed in Barzilai and Shamir [4] (page 11) for polynomially scaling dimension. Furthermore, Cao et al. [9] (Theorem 4.3) shows that this holds for Neural Tangent Kernel even more broadly, i.e. for all $i$ and not only $i \le k_m$.

# D   Sharp bounds on the eigenvalues of the Gaussian kernel

In this section, we will summarize and prove the results about eigenvalues of Gaussian and Laplace kernels when $\mathcal{X} \sim \text{Unif}(\mathbb{S}^{d-1})$.

Given a positive semi-definite kernel function $K : \mathcal{X} \times \mathcal{X} \to \mathbb{R}$, we can decompose it as

$$K(x,t) = \sum_{i=1}^{\infty} \lambda_k \phi_k(x)\phi_k(t),$$

where $\lambda_k$ and $\phi_k$ are the eigenvalues and eigenfunctions of the operator associated to $K$, $L_K : L^2_{\mathcal{D}_\mathcal{X}}(\mathbb{S}^{d-1}) \to L^2_{\mathcal{D}_\mathcal{X}}(\mathbb{S}^{d-1})$

$$L_K(f)(x) = \int_X K(x,t)f(t)d\mu(t).$$

$L_K$ is a Hilbert-Schmidt operator and has a countable system of non-negative eigenvalues $\lambda_k$ satisfying $\sum_{k=1}^{\infty} \lambda_k^2 < \infty$. The corresponding $L^2_{\mathcal{D}_\mathcal{X}}(\mathbb{S}^{d-1})$-normalized eigenfunctions $\{\phi_k(x)\}_{k=1}^{\infty}$ form an orthonormal basis of $L^2_{\mathcal{D}_\mathcal{X}}(\mathbb{S}^{d-1})$. The eigenfunctions in this case are given by spherical harmonics, $\mathcal{Y}_{i,s}$. The eigenvalues corresponding to all $\mathcal{Y}_{i,s}$ for fixed $i$ have the same eigenvalue, $\tilde{\lambda}_i$. Eigenvalues $\tilde{\lambda}_i$ are reach repeated $N(d,i) = \frac{(2i+d-2)(i+d-3)!}{i!(d-2)!}$ times. So the sequence $\{\lambda_i\}_{i=1}^{\infty} = \tilde{\lambda}_1, \ldots, \tilde{\lambda}_1, \tilde{\lambda}_2, \ldots, \tilde{\lambda}_2, \ldots$. Using Funk-Hecke formula, we can compute $\tilde{\lambda}_k$ explicitly, both for the case of Gaussian [46] kernel. For Gaussian kernel, we have the following theorem about the eigenvalues $\tilde{\lambda}_k$.

**Theorem 25** (Eigenvalues of the Gaussian kernel [46]). Let $\mathcal{X} \sim \text{Unif}(\mathbb{S}^{d-1})$, with $d \in \mathbb{N}$ and $d \ge 2$. For $K(x,t) = \exp\left(-\frac{\|x-t\|^2}{\tau_m^2}\right)$, $\tau_m > 0$, and $k \in \mathbb{N}_0$ we have that

$$\tilde{\lambda}_k = e^{-\frac{2}{\tau_m^2}}\tau_m^{d-2}I_{k+\frac{d}{2}-1}\left(\frac{2}{\tau_m^2}\right)\Gamma\left(\frac{d}{2}\right).$$

Each $\tilde{\lambda}_k$ occurs with multiplicity $N(d,k) = \frac{(2k+d-2)(k+d-3)!}{k!(d-2)!}$ (we use $\tilde{\lambda}$ notation to indicate that it has multiplicity) and its corresponding eigenfunction are the spherical harmonics of order $k$ on $\mathbb{S}^{d-1}$. Here, $I_v(z), v, z \in \mathbb{C}$ is the modified Bessel function of the first kind

$$I_v(z) = \sum_{j=0}^{\infty} \frac{1}{j!\Gamma(v+j+1)}\left(\frac{z}{2}\right)^{v+2j}.$$

It will be useful to know the size of $N(d, i)$. Also, the size of the sum of the first $k$ multiplicities will be useful. Let $N_k = \sum_{i=1}^{k} N(d, i)$.

**Lemma 26** (Size of multiplicity). For $N(d, i) = \frac{(2k+d-2)(k+d-3)!}{k!(d-2)!}$, it holds that

$$\frac{1}{(d-2)!} k^{d-2} \leq N(d, i) \leq 2^{d-1} k^{d-2}.$$

Additionally, there exist constants $n_l, n_u > 0$ depending on the dimension $d$ such that $N_k$ is bounded below and above by the following

$$n_l k^{d-1} \leq N_k \leq n_u k^{d-1}.$$

*Proof.* Since $(2k + j) \leq (2k)j$ for $j \geq 2$ and $(k + 1) \leq 2k$, we have that

$$N(d, k) = \frac{(2k + d - 2)(k + d - 3) \ldots (k + 1)}{(d - 2)!} \leq \frac{(d - 2)!(2k)^{d-3}(2k)}{(d - 2)!} \leq 2^{d-1} k^{d-2}$$

and

$$N(d, k) = \frac{(2k + d - 2)(k + d - 3) \ldots (k + 1)}{(d - 2)!} \geq \frac{1}{(d - 2)!} k^{d-2}.$$

Note that by Bernoulli's formula

$$\sum_{l=1}^{k} l^{d-2} = \frac{1}{d - 1} \left( k^{d-1} + o(k^{d-1}) \right).$$

Therefore, we have that

$$n_l k^{d-1} \leq \frac{1}{(d - 1)(d - 2)!} \left( k^{d-1} + o(k^{d-1}) \right) \leq N_k \leq \frac{2^{d-1}}{d - 1} \left( k^{d-1} + o(k^{d-1}) \right) \leq n_u k^{d-1}.$$

$\square$

**Lemma 27** (Inverting the index). If the index of an eigenvalue $\hat{\lambda}_j$ is such that $N_{k-1} \leq j \leq N_k - 1$, then $\hat{\lambda}_j = \lambda_k$. We will denote such $j$ with $\tilde{k}$, i.e. $\tilde{k}$ is an index such that $\hat{\lambda}_{\tilde{k}} = \lambda_k$.

*Proof.* Immediate from the fact that $\{\lambda_i\}_{i=1}^{\infty}$ is a sequence with $\tilde{\lambda}_i$ repeating $N(d, i)$ times, in that order. $\square$

An interesting property of eigenvalues of the Gaussian kernel is that they are sorted because the Modified Bessel functions are [48]. In particular, $I_{v+1}(x) < I_v(x)$ for all $v, x > 0$, so $\tilde{\lambda}_{i+1} < \tilde{\lambda}_i$. This is not the case for the Laplace kernel.

**Theorem 28** (Size of Ratios of Eigenvalues for Gaussian Kernel). Let $T = (\frac{2}{\tau_m^2})$. For $j \leq \sqrt{T}t(m), k \leq \sqrt{T}t(m)$, where $t(m) \to 0$ as $m \to \infty$, we have that

$$\exp(-6t(m)) < \frac{\tilde{\lambda}_{k+j}}{\tilde{\lambda}_k} < 1.$$

For $j \geq \sqrt{T}t(m)$, where $t(m) \to \infty$ as $m \to \infty$ we have that for any $k$

$$\frac{\tilde{\lambda}_{k+j}}{\tilde{\lambda}_k} < \exp\left(-\frac{t(m)^2}{4}\right).$$

*Proof.* First of all, note that from [48], we have that for $v \geq \frac{1}{2}$

$$\frac{(v) + \sqrt{(v)^2 + x^2}}{x} > \frac{I_{v-1}(x)}{I_v(x)} > \frac{(v - \frac{1}{2}) + \sqrt{(v - \frac{1}{2})^2 + x^2}}{x}$$

$$\frac{x}{(v) + \sqrt{(v)^2 + x^2}} < \frac{I_v(x)}{I_{v-1}(x)} < \frac{x}{(v - \frac{1}{2}) + \sqrt{(v - \frac{1}{2})^2 + x^2}}$$

In particular, this means that for the eigenvalues $\tilde{\lambda}_k$, we have

$$\frac{(\frac{2}{\tau_m^2})}{(k + \frac{d}{2}) + \sqrt{(k + \frac{d}{2})^2 + (\frac{2}{\tau_m^2})^2}} < \frac{\tilde{\lambda}_{k+1}}{\tilde{\lambda}_k} = \frac{I_{k+1+\frac{d}{2}-1}(\frac{2}{\tau_m^2})}{I_{k+\frac{d}{2}-1}(\frac{2}{\tau_m^2})} < \frac{(\frac{2}{\tau_m^2})}{(k + \frac{d}{2} - \frac{1}{2}) + \sqrt{(k + \frac{d}{2} - \frac{1}{2})^2 + (\frac{2}{\tau_m^2})^2}}.$$

This can be bounded with a simpler expression as follows

$$\frac{(\frac{2}{\tau_m^2})}{2(k + \frac{d}{2}) + (\frac{2}{\tau_m^2})} < \frac{\tilde{\lambda}_{k+1}}{\tilde{\lambda}_k} = \frac{I_{k+1+\frac{d}{2}-1}(\frac{2}{\tau_m^2})}{I_{k+\frac{d}{2}-1}(\frac{2}{\tau_m^2})} < \frac{(\frac{2}{\tau_m^2})}{(k + \frac{d}{2} - \frac{1}{2}) + (\frac{2}{\tau_m^2})}.$$

We will use these bounds to derive tight bounds for the ratios $\frac{\tilde{\lambda}_{k+j}}{\tilde{\lambda}_k}$. Note the following

$$\prod_{i=1}^{j} \frac{(\frac{2}{\tau_m^2})}{2(k + i - 1 + \frac{d}{2}) + (\frac{2}{\tau_m^2})} < \frac{\tilde{\lambda}_{k+j}}{\tilde{\lambda}_k} = \prod_{i=1}^{j} \frac{\tilde{\lambda}_{k+i}}{\tilde{\lambda}_{k+i-1}} < \prod_{i=1}^{j} \frac{(\frac{2}{\tau_m^2})}{(k + i - 1 + \frac{d}{2} - \frac{1}{2}) + (\frac{2}{\tau_m^2})}.$$

Note now that

$$\left( \frac{(\frac{2}{\tau_m^2})}{2(k + j - 1 + \frac{d}{2}) + (\frac{2}{\tau_m^2})} \right)^j < \prod_{i=1}^{j} \frac{(\frac{2}{\tau_m^2})}{2(k + i - 1 + \frac{d}{2}) + (\frac{2}{\tau_m^2})} < \frac{\tilde{\lambda}_{k+j}}{\tilde{\lambda}_k}.$$

Note that since $(x + j - i)(x + i) = x^2 + ix + (j - i)i$, we have that $(x + j - i)(x + i) \geq (x + j)x$, therefore

$$\prod_{i=1}^{j} \frac{(\frac{2}{\tau_m^2})}{(k + i - 1 + \frac{d}{2} - \frac{1}{2}) + (\frac{2}{\tau_m^2})}$$

$$< \left( \frac{(\frac{2}{\tau_m^2})}{(k + j - 1 + \frac{d}{2} - \frac{1}{2}) + (\frac{2}{\tau_m^2})} \right)^{\frac{j-1}{2}} \left( \frac{(\frac{2}{\tau_m^2})}{(k + \frac{d}{2} - \frac{1}{2}) + (\frac{2}{\tau_m^2})} \right)^{\frac{j+1}{2}}.$$

We use $j + 1$ and $j - 1$ to account for the fact that $j$ might be odd when we split into $(j - 1)/2$ pairs. When $j$ is even, we split into $\frac{j}{2}$ pairs and use the fact that the term with exponent $\frac{j+1}{2}$ is larger.

Let $T = (\frac{2}{\tau_m^2})$. We can bound the ratio $\frac{\tilde{\lambda}_{k+j}}{\tilde{\lambda}_k}$ tightly now.

When $j \leq \sqrt{T}m^{-\delta}, k \leq \sqrt{T}m^{-\delta}$, we have that

$$\exp(-6m^{-\delta}) < \frac{\tilde{\lambda}_{k+j}}{\tilde{\lambda}_k} < 1.$$

When $j \geq \sqrt{T}m^\delta$, we have that for any $k$

$$\frac{\tilde{\lambda}_{k+j}}{\lambda_k} < \exp(-\frac{m^{2\delta}}{4}).$$

To see why this is true, note first that

$$\left(\frac{(\frac{2}{\tau_m^2})}{2(k+j-1+\frac{d}{2})+(\frac{2}{\tau_m^2})}\right)^j$$

$$= \left(\frac{T}{2(k+j-1+\frac{d}{2})+T}\right)^j > \left(\frac{T}{2(k+j-1+\frac{d}{2})+T}\right)^{\sqrt{T}}$$

$$> \left(\frac{T}{5(\sqrt{T}m^{-\delta})+T}\right)^{\sqrt{T}} = \left(\frac{1}{5(\frac{1}{\sqrt{T}}m^{-\delta})+1}\right)^{\sqrt{T}} \to \exp(-5m^{-\delta}),$$

as $m \to \infty$. For the second inequality, note that

$$\left(\frac{(\frac{2}{\tau_m^2})}{(k+\frac{d}{2}-\frac{1}{2})+(\frac{2}{\tau_m^2})}\right)^{\frac{j+1}{2}} < 1.$$

We also have that

$$\left(\frac{(\frac{2}{\tau_m^2})}{(k+j-1+\frac{d}{2}-\frac{1}{2})+(\frac{2}{\tau_m^2})}\right)^{\frac{j-1}{2}} = \left(\frac{T}{(k+j-1+\frac{d}{2}-\frac{1}{2})+T}\right)^{\frac{j-1}{2}} < \left(\frac{T}{(\sqrt{T}m^\delta)+T}\right)^{\frac{\sqrt{T}m^\delta-1}{2}}$$

$$= \left(\frac{1}{\frac{(\sqrt{T}m^\delta)}{T}+1}\right)^{\frac{\sqrt{T}m^\delta}{3}} < \left(\frac{1}{\frac{m^\delta}{\sqrt{T}}+1}\right)^{\frac{\sqrt{T}m^\delta}{3}} = \left(\left(\frac{1}{\frac{m^\delta}{\sqrt{T}}+1}\right)^{\frac{\sqrt{T}}{m^\delta}}\right)^{\frac{m^{2\delta}}{3}}$$

$$\to \exp(-\frac{m^{2\delta}}{3}) \to 0,$$

as $m \to \infty$. Note that we can turn the limits $a_m \to t(m)$ into inequalities of the form $(1-\varepsilon)t(m) < a_m < (1+\varepsilon)t(m)$ for some $\varepsilon$ and all $m$ since the convergence is uniform. $\square$

The following is a simple corollary.

**Corollary 29.** Let $T = (\frac{2}{\tau_m^2})$. If $T > 1$ and the index of the eigenvalue $i$ is such that $i \leq (k\sqrt{T})^{d-1}$, we have that for $\lambda_i$ it holds that

$$\frac{\lambda_i}{\lambda_1} \geq \left(\frac{1}{1+\frac{k}{\sqrt{T}}}\right)^{k\sqrt{T}} \geq \exp(-k^2).$$

*Proof.* Note that $\left(\frac{1}{1+\frac{k}{\sqrt{T}}}\right)^{k\sqrt{T}}$ is increasing in $\sqrt{T}$ and note that $\sqrt{T}$ increases as $m$ increases. Note that for $\sqrt{T} = 1$ it suffices to show $e(k) > 1 + k$ which is true for all $k$ as long as $\sqrt{T} > 1$. $\square$

The following simple bound also holds for eigenvalues of Gaussian kernel.

**Proposition 30** (Ratio of eigenvalues bounded above [46]). For the eigenvalues associated to the Gaussian Kernel $\tilde{\lambda}_k$ of bandwidth $\tau_m$ we have that

$$\frac{\tilde{\lambda}_{k+1}}{\tilde{\lambda}_k} < \frac{1}{\tau_m^2(k+\frac{d}{2})}.$$

It is straightforward to convert the bounds on ratios of eigenvalues Theorem 25 to bounds on the sizes of the actual eigenvalues.

**Corollary 31** (Sizes of the eigenvalues of Gaussian kernel). The following bounds hold for the largest eigenvalue of the Gaussian kernel

$$\frac{1}{\tau_m^2 + 4} \frac{\Gamma((\frac{2}{\tau_m^2} + \frac{1}{2}))\Gamma(\frac{d}{2})}{\Gamma(\frac{2}{\tau_m^2} + \frac{d}{2} + 2)} < \tilde{\lambda}_1 < \frac{1}{\sqrt{\tau_m^4 + 4\tau_m^2}} \frac{\Gamma((\frac{2}{\tau_m^2} + \frac{1}{2}))\Gamma(\frac{d}{2})}{\Gamma(\frac{2}{\tau_m^2} + \frac{d}{2} + \frac{3}{2})}.$$

Therefore, for $\tilde{\lambda}_k$, we have that

$$\frac{1}{\tau_m^{2k}} \frac{\Gamma((\frac{2}{\tau_m^2} + k + \frac{1}{2}))}{\Gamma(\frac{2}{\tau_m^2} + k + \frac{d}{2} + 2)} \tilde{\lambda}_1 < \tilde{\lambda}_k$$

$$\tilde{\lambda}_k < 2^k \frac{1}{\tau_m^{2k}} \frac{\Gamma((\frac{2}{\tau_m^2} + k + \frac{1}{2}))}{\Gamma(\frac{2}{\tau_m^2} + k + \frac{d}{2} + \frac{3}{2})} \tilde{\lambda}_1.$$

Furthermore, for $\tau_m$ fixed we have that

$$\tilde{\lambda}_1 > \frac{1}{\tau_m^2 + 4} \frac{1}{((\frac{d}{2} + \frac{2}{\tau_m^2})^{1 + \frac{2}{\tau_m^2}})}.$$

and

$$\tilde{\lambda}_k > \frac{1}{\tau_m^2 + 4} \frac{1}{((\frac{d}{2} + \frac{2}{\tau_m^2})^{1 + \frac{2}{\tau_m^2}})} \frac{1}{\tau_m^{2k}} \frac{1}{((k + \frac{d}{2} + \frac{2}{\tau_m^2})^k)}.$$

*Proof.* Note that from [62] we have

$$\frac{1}{1 + \frac{4}{\tau_m^2}} \exp\left(\frac{2}{\tau_m^2}\right) < I_0\left(\frac{2}{\tau_m^2}\right) < \frac{1}{\sqrt{1 + \frac{4}{\tau_m^2}}} \exp\left(\frac{2}{\tau_m^2}\right)$$

We also have from [48]

$$\frac{(\frac{2}{\tau_m^2})}{2(i+1) + (\frac{2}{\tau_m^2})} < \frac{I_{i+1}(\frac{2}{\tau_m^2})}{I_i(\frac{2}{\tau_m^2})} < \frac{(\frac{2}{\tau_m^2})}{2(i + \frac{1}{2})}.$$

Then

$$\tilde{\lambda}_k = e^{-\frac{2}{\tau_m^2}} \tau_m^{d-2} I_{k + \frac{d}{2} - 1}\left(\frac{2}{\tau_m^2}\right) \Gamma\left(\frac{d}{2}\right).$$

So then

$$\frac{1}{\tau_m^d} \frac{\Gamma((\frac{2}{\tau_m^2} + \frac{1}{2}))}{\Gamma(\frac{2}{\tau_m^2} + \frac{d}{2} + 2)} < \prod_{i=0}^{\frac{d}{2}-1} \frac{(\frac{2}{\tau_m^2})}{2(i+1) + (\frac{2}{\tau_m^2})} < \frac{I_{1 + \frac{d}{2} - 1}\left(\frac{2}{\tau_m^2}\right)}{I_0\left(\frac{2}{\tau_m^2}\right)} < \prod_{i=0}^{\frac{d}{2}-1} \frac{(\frac{2}{\tau_m^2})}{2(i + \frac{1}{2})} = \frac{1}{\tau_m^d} \frac{\Gamma(\frac{1}{2})}{\Gamma(\frac{d}{2})}.$$

So then

$$\frac{1}{\tau_m^2 + 4} \frac{\Gamma((\frac{2}{\tau_m^2} + \frac{1}{2}))\Gamma(\frac{d}{2})}{\Gamma(\frac{2}{\tau_m^2} + \frac{d}{2} + 2)} < \tilde{\lambda}_1 < \frac{1}{\sqrt{\tau_m^4 + 4\tau_m^2}} \frac{\Gamma((\frac{2}{\tau_m^2} + \frac{1}{2}))\Gamma(\frac{d}{2})}{\Gamma(\frac{2}{\tau_m^2} + \frac{d}{2} + \frac{3}{2})}.$$

By repeating the same argument, the claim about $\tilde{\lambda}_k$ follows.

Note that

$$\tilde{\lambda}_1 > \frac{1}{\tau_m^2 + 4} \Gamma(\frac{d}{2}) \prod_{i=0}^{\frac{d}{2}-1} \frac{1}{(i+1) + (\frac{2}{\tau_m^2})} > \frac{1}{\tau_m^2 + 4} \frac{1}{((\frac{d}{2} + \frac{2}{\tau_m^2})^{1+\frac{2}{\tau_m^2}})}.$$

Therefore, we have that

$$\tilde{\lambda}_k > \tilde{\lambda}_1 \prod_{i=0}^{k-1} \frac{\frac{2}{\tau_m^2}}{k + \frac{d}{2} + (\frac{2}{\tau_m^2})} > \tilde{\lambda}_1 \frac{1}{\tau_m^{2k}} \frac{1}{((k + \frac{d}{2} + \frac{2}{\tau_m^2})^k)}.$$

$\square$

