# OpenReview forum: "Overfitting Behaviour of Gaussian Kernel Ridgeless Regression: Varying Bandwidth or Dimensionality"
_NeurIPS.cc/2024/Conference — NeurIPS 2024 poster_

### Official Review · Reviewer_zGqe · 2024-07-04

**Soundness:** 2
**Presentation:** 3
**Contribution:** 1
**Rating:** 3
**Confidence:** 4

**Summary:**

This paper concerns the overfitting behaviour of Gaussian kernel ridge regression (KRR) with varying bandwidth or dimensionality. The contribution is two-fold:

1. In fixed-dimension, the ridgeless solution of Gaussian KRR is not consistent with any varying or tuned bandwidth.

2. In high dimension with input dimension $d=2^{2^l}$, sample size $m=2^{2^{2^l}}$ for any arbitrary integer $l$, the overfitting of Gaussian KRR is benign.

All the results are under the Gaussian universality Ansatz and the (non-rigorous) risk predictions in terms of the kernel eigenstructure introduced in [Simon2023].

Reference:

- James B. Simon, Madeline Dickens, Dhruva Karkada, and Michael R. DeWeese. The eigenlearning framework: A conservation law perspective on kernel regression and wide neural networks. arXiv preprint arXiv:2110.03922, 2021.

**Strengths:**

The two contributions are novel.

**Weaknesses:**

However, I have serious concerns about the assumptions made in this paper. Firstly, the eigenframework under the Gaussian Universality Ansatz (GUA) is non-rigorous and actually fails to explain the catastrophic overfitting of NTK in fixed dimensions, as reported in \cite{Barzilai2023} and \cite{Cheng2024}. I am uncertain whether benign overfitting would still hold in high-dimensional settings. This paper does not offer experimental validation either, so I cannot ensure the correctness of the claim for Gaussian kernels (not its proxy under GUA).

Secondly, the catastrophic overfitting of Gaussian kernels with fixed bandwidth is well-known. It is not difficult to imagine that Gaussian KRR with varying bandwidth remains inconsistent. While the first contribution of this paper is novel, I question whether it is significant enough for paper acceptance, especially given my concerns regarding the second contribution.

Reference:
- Barzilai, Daniel, and Ohad Shamir. "Generalization in kernel regression under realistic assumptions." arXiv preprint arXiv:2312.15995 (2023).
- Cheng, Tin Sum, et al. "Characterizing overfitting in kernel ridgeless regression through the eigenspectrum." arXiv preprint arXiv:2402.01297 (2024).

**Questions:**

The most important question would be if the authors could provide some experimental validations on these two claims.

**Limitations:**

All assumptions are stated in the claims. However, as mentioned before, GUA and Eigenframework assumptions are too unrealistic and might provide wrong results.

---

> ### Author Rebuttal · Authors · 2024-08-07
>
> We thank the reviewer for their feedback. Below we address the main comments:
> 1. Regarding the wrong predictions of the Gaussian Universality Ansatz (GUA): There is ample empirical evidence that the eigenframework holds well for the Gaussian kernel. For example, Figure 1 in [38] shows the predicted vs. true test risk for Gaussian KRR (with data from MNIST) and Figure 5 in [7] shows the true vs. predicted risk for Gaussian KRR with data uniform on the sphere (which is the setup we consider here). In both of these plots, the prediction matches closely with the true test risk. Furthermore, there are many works that rely on the eigenframework, such as [1, 7, 12, 25, 47, 54, 63]. Therefore, this limitation holds for the entire line of work and not just for the current paper. We agree with the reviewer that understanding the limitations of this framework is an important future research direction. We will update the paper to highlight this as a limitation.
>
> 2. Experimental results confirming the validity of our claims: In the attached PDF, we plotted the dependence of the test error on sample size for the Gaussian kernel ridgeless (interpolating) predictor. In Figure a, we consider the first case of Theorem 1, namely when the bandwidth $\tau_m = o(m^{-\frac{1}{d-1}})$. It shows that the test error tends to something equal to or larger than the test risk of the null predictor. In Figure b, we consider the second case, i.e $\tau_m = \omega(m^{-\frac{1}{d-1}})$ and the plot shows that the test error increases with sample size, which aligns with the claim that it diverges as the sample size $m$ increases. In Figure c, we consider the case when $\tau_m = \Theta(m^{-\frac{1}{d-1}})$ and the plot shows that for some noise level (which is high enough), the predictor is worse than the null predictor, as predicted by our theorem. We will add these simulations to the final version of the paper.

---

> > ### Comment · Reviewer_zGqe · 2024-08-09
> >
> > Thank you for your prompt response and the experimental results. We can now be confident that the Gaussian kernels behave as predicted by the theory. However, for a theoretical result, I would expect a more realistic assumption to explain the Gaussian kernel interpolation. As mentioned by other reviewers, the major technique was developed in earlier works, and the results presented in this paper alone do not justify acceptance. I believe one possible direction could be to drop the GUA assumption in your analysis.

---

> > > ### Author Response · Authors · 2024-08-11
> > >
> > > We thank the reviewer for their further feedback. We believe that the discussion about the significance of the theoretical results is important for all the reviewers to see, so we are providing a global reply to this discussion.

---

### Official Review · Reviewer_dZGD · 2024-07-07

**Soundness:** 2
**Presentation:** 2
**Contribution:** 2
**Rating:** 5
**Confidence:** 4

**Summary:**

This paper discusses the generalization ability of Gaussian kernel interpolation, an interesting topic. However, the authors sidestep the most challenging part by assuming the 'Gaussian universality ansatz,' which significantly simplifies the problem. With this assumption, their work essentially consists of elementary computations. Furthermore, their argument heavily relies on the earlier work by Lijia Zhou, James B. Simon, Gal Vardi, and Nathan Srebro, titled 'An agnostic view on the cost of overfitting in 473 (kernel) ridge regression,' an arXiv preprint with the identifier arXiv:2306.13185, published in 2023."

**Strengths:**

A detailed calculation of the eigenvalue structure of the Gaussian kernel with various bandwidths has been performed. This could be valuable for future studies.

**Weaknesses:**

No significant contribution was made to the kernel interpolation problem itself. The major workhorse was developed in earlier work by Lijia Zhou, James B. Simon, Gal Vardi, and Nathan Srebro, as detailed in their paper titled 'An agnostic view on the cost of overfitting in 473 (kernel) ridge regression,' published as an arXiv preprint with the identifier arXiv:2306.13185 in 2023.

**Questions:**

Perhaps due to my lack of knowledge, could the author elaborate on their contributions beyond the detailed computation of the Gaussian kernel with varying bandwidths?

**Limitations:**

None.

---

> ### Author Rebuttal · Authors · 2024-08-06
>
> We thank the reviewer for their feedback. Below we address the main comments:
>
> 1. The goal of the paper is to study the kernel interpolation problem by understanding its behavior under common settings. It achieves this goal in two ways. First, we consider the common Gaussian kernel with varying bandwidth. In lines 82-86, we argue that this is an important setting since it achieves optimal rates of convergence for a large set of target functions (note that the optimal rates are achieved by varying the bandwidth). Indeed, using a varying bandwidth is a standard practice when using the Gaussian kernel. We are the first to consider whether the interpolating solution exhibits tempered or catastrophic behavior in this setting and how it compares to the null predictor. This result shows that even with varying bandwidth, the interpolating solution is almost always worse than the null predictor. Second, we consider increasing dimensions and sample size with more general kernels. We establish upper and lower bounds on the test risk for any scaling of the dimension and sample size, which indicates what type of overfitting we get. This result not only agrees with the known case of polynomial scaling of the dimension but also allows us to show the first case of sub-polynomially increasing dimension that achieves benign overfitting for the Gaussian kernel. Our results provide a more comprehensive analysis of overfitting in kernel interpolation.
>
>
> 2. The results are highly non-trivial at the technical level and do not simply use Zhou et al 2023 [63] (as the reviewer suggested). First, for fixed dimension, our technical contribution is deriving the dependence of the eigenvalues of the Gaussian kernel on the bandwidth (Theorem 8, line 1133) and hence understanding how the test risk depends on the bandwidth. Second, for increasing dimension, our technical contribution is deriving upper and lower bounds for the test risk that work for any scaling of the dimension and sample size (Theorems 2 and 3, lines 189 and 203), by introducing the “upper and lower index” (Definition 1, line 182).

---

> ### Comment · Reviewer_dZGD · 2024-08-10
>
> In kernel regression, there are two quantities that characterize the generalization ability of the kernel estimator (and for interpolation, there might be only one such quantity). These quantities are in turn determined by the eigenvalues of the kernel.
>
> When considering the Gaussian kernel, there is an explicit formula for the eigenvalues in terms of the bandwidth h. Thus, if
> h varies, the two quantities will vary as well. Since the author has further adopted the Gaussian ansatz, I assume the major contribution is characterizing how the eigenvalues and these two quantities vary with changes in
> h. Could you please correct me if I have overlooked any other key contributions?

---

> > ### Author Response · Authors · 2024-08-11
> >
> > We thank the reviewer for additional comments. Indeed, for the interpolating solution of the Gaussian KRR, there is one parameter that can be varied, the bandwidth $\tau_m$. Note that the bandwidth can (and in practice often does) depend on sample size $m$. Our first technical contribution, which is understanding how the eigenvalues of the Gaussian kernel for the sphere depend on the bandwidth $\tau _ m$, holds without the Gaussian design ansatz assumption. The second technical contribution is understanding how the test risk of the interpolating solution of Gaussian KRR depends on the bandwidth and thus characterizing its overfitting behavior, under the Gaussian design ansatz. Note that both of these are highly nontrivial technical contributions. Our third technical contribution is for the case of increasing dimension, understanding how the test risk behaves for any scaling of the dimension and sample size. We achieve this by showing upper and lower bounds on the test risk of the interpolating solution of KRR. Our fourth technical contribution is using these bounds for increasing dimension to demonstrate that we can achieve benign overfitting even with sub-polynomially increasing dimension and Gaussian KRR, improving the currently known result for polynomially increasing dimension. This is also a nontrivial technical contribution.

---

> > > ### Comment · Reviewer_dZGD · 2024-08-11
> > >
> > > Thanks for your clarification. I have increased the score.

---

### Official Review · Reviewer_FXXZ · 2024-07-09

**Soundness:** 2
**Presentation:** 2
**Contribution:** 2
**Rating:** 6
**Confidence:** 4

**Summary:**

This paper looks at the problem of kernel (ridgeless) regression with Gaussian kernel and data on a sphere. The paper studies two cases, first we fix the dimension $d$ and are allowed to send the number of data points $m \to \infty$ and the width of the kernel $\tau_m \to 0$, second, we are allowed to scale $d$. For both problems they study how a quantity known as the predicted risk scales. *It is important to note that the paper assumes that this is a good estimate of the true excess risk, which is not immediately evident and I think requires more justification (see weaknesses).*

In the first case they show that if $\tau_m$ goes to zero too quickly, we have tempered overfitting for the *preditcted risk* and that it has a worse risk than the null estimator. If it goes to 0 too slowly, then we have catastrophic overfitting for the *predicted risk*, and that if it goes to zero at the appropriate rate, we have tempered overfitting again and that we may outperform the null estimator. The paper also presents some results about the case when $d$ scales as well.

**Strengths:**

The paper uses the predicted risk as an estimate of the true risk. The prdicted risk is given by

$$ \tilde{R}(\tilde{f}\_{\delta})  = \mathcal{E}\_\delta \left(\sum\_{i=1}^\infty  (1-\mathcal{L}\_{i,\delta})^2 \beta_i^2 + \sigma^2 \right) $$

Where $\beta_i$ are the coefficients of the target function in the eigenbasis of the kernel function and $\mathcal{E}, \mathcal{L}$ are coefficients based on eigenvalues and $\sigma^2$ is the noise variance. *If we assume that this is a good estimate of the excess risk*. Then to understand the risk, we have reduced the problem to understanding how $\mathcal{E}, \mathcal{L}$ **and** $\beta_i$ scale.

**This paper studies how $\mathcal{E}_0$ scales by providing some bounds on the eigenvalues of the Gaussian kernel function. This I think is the primary contribution of the paper.**

However, there are some issues, see weaknesses.

**Weaknesses:**

There are a few weaknesses. The current reject score is due to my concerns with the proofs.

1) **Theorem 1**

I have some concerns with the results in Theorem 1. Theorem 1 states that unless we meet some very specific conditions, the null predictor (i.e. the zero function) outperforms kernel ridge regression. This seems quite surprising to me.

Taking a closer look at the proof, we see that proof is broken down into two Lemmas. One proves the upper bounds, the other lower bounds. However, Lemma 1, only deals with $\mathcal{E}_0$. However, I think something has to be said about $\sum\_{i=1}^\infty  (1-\mathcal{L}\_{i,\delta})^2 \beta_i^2$.

If I understand correctly, when we change $m$, we change the width, which changes the kernel function. However, since $\lambda_i$ and $\phi_i$ are the dependent on $K$. Hence $\beta_i$ which are the coefficients of $f$ in this eigenbasis are dependent on $m$. Hence $\sum\_{i=1}^\infty  (1-\mathcal{L}\_{i,\delta})^2 \beta_i^2$ has a subtle dependence on $m$ which I think needs to be accounted for. Because apriori it is not clear to me that this is bounded in $m$. That is, $\sum_i \beta_i^2(m) = S(m)$ could be unbounded in $m$.

I did not go through the lower bound proofs as carefully, but I do think the the dependence of $\beta_i$ on $m$ is addresses. Please correct me if I am wrong.

Hence I think either another assumption needs to be added (and if it is added a discussion of its reasonableness would be appreciated) or that terms needs to be explicity dealt with.

2) Eigenlearding Framework and Predicted Risk

In general, I would prefer results showing that the predicted risk is a good estimate of the true risk. The paper on lines 116-131 attempt to this. However, the primary papers that have been cited [1,12,25] are all about **linear** regression and not **kernel** regression. Specifically, they all assume that $y_i = \beta^T x_i + \epsilon_i$ and that we are fitting a linear function. This is not quite the setting of this paper, as we have an infinite dimensional feature vector space that we are working in. Hence I do not think those papers are sufficient theoretical justification.

My other concern with this, is that the discussion from lines 98-111 **do not assume that** $y = f(x) + \epsilon_i$ which I think is needed in [47]. The paper presents for general distributiuons on $\mathcal{X}$ and $\mathcal{Y}$. I do not think the eigenlearning framework captures the following situation $x_i  = z_i + \epsilon_j$ and $y_i = f(z_i)$, but the paper here, as written, would claim this. See [A,B] for papers that address such a situation, [B] even has results on benign overfitting.


[A] Sonthalia, Rishi and Raj Rao Nadakuditi. “Training Data Size Induced Double Descent For Denoising Feedforward Neural Networks and the Role of Training Noise.” Trans. Mach. Learn. Res. 2023 (2023): n. pag.

[B] Kausik, C., Srivastava, K., & Sonthalia, R. (2023). Double Descent and Overfitting under Noisy Inputs and Distribution Shift for Linear Denoisers.TMLR 2024

3) References

I think the long list of references on lines 45-48 need to be changed. The paper should either further subcategorize them, or cite fewer papers. In the case that the authors decide to further subvategorize them. I think they are missing some references [A,B,C,D]. Additionally, some papers are cited twice, see 25, 26, and 23, 24.

[C] Xiao, L., Hu, H., Misiakiewicz, T., Lu, Y., & Pennington, J. (2022). Precise Learning Curves and Higher-Order Scalings for Dot-product Kernel Regression. Neural Information Processing Systems.

[D] Hu, H., Lu, Y.M., & Misiakiewicz, T. (2024). Asymptotics of Random Feature Regression Beyond the Linear Scaling Regime. ArXiv, abs/2403.08160.

**Questions:**

Please see weaknesses

**Limitations:**

The limitations are well discussed.

---

> ### Author Rebuttal · Authors · 2024-08-06
>
> We thank the reviewer for the thoughtful feedback. Below we address the main comments:
>
> 1. Regarding the claim about Theorem 1:
> $\beta_i$ in this case do not depend on $m$. Changing $m$ does change the bandwidth $\tau_m$, but the eigenfunctions remain the same despite the kernel changing - the eigenfunctions are the spherical harmonics. Therefore, $\beta_i$ does not depend on $m$, as it is $\beta_i = \mathbb E_{D_X} \left( f^* \phi_{i}\right)$ and $\phi_{i}$ is a fixed spherical harmonic in $d$ dimensions (i.e. $Y_{ks}$ for some $k$ and $s$; here $d$ is fixed). Therefore $\sum_i \beta_i^2 = S(m)$ is fixed and independent of $m$.
> Note however that $\mathcal{L}_{i,\delta}$ does depend on $m$ as it depends on the eigenvalues and this is addressed in both the upper and lower bounds (although indirectly for the lower bound).
>
>
> 2. Regarding the eigenlearning framework: In lines 126-129, we discuss a more recent work, [38], in which they provide theoretical justification for eigenframework in the case of kernel regression. Since there are quite a few theoretical works that rely on the eigenlearning framework (including ours), for example [1, 7, 12, 25, 47, 54, 63],  and since there is vast empirical evidence to support it, we agree with the reviewer that obtaining further theoretical justification for it is an important topic for future research.
> The assumption on the target in 98-111: It’s true that as it’s written, it is not yet assumed that $y = f^*(x) +\xi$. However, as discussed in Zhou et al 2023 [63], the results of [47] can be extended to a more general setup. We will edit the paper to clarify this.
>
> 3. Regarding the references: Thanks. We will update this section of the paper to include these additional references.

---

> > ### Comment · Reviewer_FXXZ · 2024-08-07
> >
> > Could the authors expand on how we know the eigenfunctions are spherical harmonics? Even a reference to another work would be appreciated.
> >
> > Line 100 seems to cite Mercer's Theorem. These notes (https://www.stat.berkeley.edu/~bartlett/courses/2014spring-cs281bstat241b/lectures/20-notes.pdf) by Peter Bartlett seem like a reasonable reference to me. Here the order of implications is for a given kernel there exist eigenfunctions such that...  so it looks like the eigenfunctions are dependent on the kernel.

---

> > > ### Author Response · Authors · 2024-08-07
> > > **Reply**
> > >
> > > Yes, we can provide references and explain further. In [43], in section 2 and Theorem 2, it is explained that when we are on $S^{d-1}$, the eigenfunctions of the Gaussian kernel are spherical harmonics. This is shown in their proof of Theorem 2 in equation (24). Namely, using the Funk-Hecke formula (see page 30 of the reference below), one can show that for any inner product kernel on the sphere, the spherical harmonics are the eigenfunctions. The Funk-Hecke formula states that for any spherical harmonic of order $k$, $Y_k$, and any $x\in S^{d-1}$ the following equation holds $\int_{S^{d-1}} f(\langle x,t \rangle) Y_k(t) dS^{d-1}(t) = \lambda_k Y _ k(x)$, where $\lambda_k$ is a real number given by an integral that depends on $f$ (see [43] for the formula). Then, if our kernel $K: X\times X\to \mathbb R$ is an inner product kernel, there is a function $k: \mathbb R\to \mathbb R$ such that $K(x,t) = k(\langle x,t\rangle)$. Then applying the Funk-Hecke formula for $f=k$ would give the desired result. Note that on $S^{d-1}$, we have that $\exp\left(-\frac{\| x-t \|^2}{\sigma^2} \right) = \exp \left( -\frac{2}{\sigma^2}\right) \exp \left( \frac{2 \langle x,t \rangle }{\sigma^2} \right)$, which shows that the Gaussian kernel is an inner product kernel. For the proof of the Funk-Hecke formula, the reviewer can look at the reference given below. The fact that eigenfunctions of inner product kernels are spherical harmonics is frequently used in papers that discuss inner product kernels, such as [4] (Section G).
> > >
> > > Funk-Hecke formula reference: C. Muller. Analysis of Spherical Symmetries in Euclidean Spaces. Applied Mathematical Sciences 129, Springer, New York, 1997.
> > >
> > >
> > > [43] Minh Ha Quang and Yuan Yao. Mercer’s theorem, feature maps, and smoothing. Conference: Learning Theory, 19th Annual Conference on Learning Theory, COLT, 2006.
> > >
> > >
> > > [4] Daniel Barzilai and Ohad Shamir. Generalization in kernel regression under realistic assumptions. arXiv preprint arXiv:2312.15995v2, 2024.

---

> > > > ### Comment · Reviewer_FXXZ · 2024-08-10
> > > >
> > > > Thank you for the clarification. I have increased my score

---

> > > > > ### Author Response · Authors · 2024-08-11
> > > > >
> > > > > We thank the reviewer for a productive discussion and for increasing our score.

---

### Official Review · Reviewer_cvRM · 2024-07-11

**Soundness:** 3
**Presentation:** 4
**Contribution:** 3
**Rating:** 7
**Confidence:** 4

**Summary:**

The paper investigates the overfitting behavior of Gaussian Kernel Ridge Regression (KRR) in high-dimensional settings, focusing on how the model performance is influenced by the choice of the bandwidth parameter and sample size.

They aimed to provide a more comprehensive picture of overfitting with Gaussian KRR by studying the overfitting behavior with varying bandwidth or with arbitrarily varying dimension, including sub-polynomially. In particular, they showed that for fixed dimension, even with varying bandwidth, the interpolation learning is never consistent and generally not better than the null predictor (either the test error tends to infinity or is finite but it is almost always not better than the null predictor). For increasing dimension, they gave an upper and lower bound on the test risk for any scaling of the dimension with sample size, which indicates in many cases whether the overfitting is catastrophic, tempered or benign. They showed the first example of sub-polynomially scaling dimension that achieves benign overfitting for the Gaussian kernel. Additionally, they showed that a class of dot-product kernels on the sphere is inconsistent when the dimension scales logarithmically with sample size.

**Strengths:**

The presentation of the paper looks really in good order with adequate clarifications and rich literature reviews. The motivation is clear to me. The proof provided also looks correct in general.,

They clearly provided an upper and lower bound on the test risk for any scaling of the dimension with sample size, which provides a clear and comprehensive understanding on the issue of overfitting with Gaussian KRR, which is not restricted on particular regimes, while it also discovers new result for sub-polynomially varying dimension. I found the proof technique quite interesting and insightful.

Although the contribution is not extremely outstanding or groundbreaking, the results completely reflect the effect of varying bandwidth or dimensionality in a very comprehensive way and it certainly facilitates the understanding of this issue and also provides a very insightful theoretical guidance of picking these crucial parameters.

**Weaknesses:**

The first and a clear weakness would be lack of experiments. Since the results in the paper focus on the overfitting behavior with varying bandwidth or with arbitrarily varying dimension, including sub-polynomially, it should benefit a lot and is very instructive to study the real effects guided by the theories on the simulated or real-word datasets.

The results should be sensitivity to tuning parameters. The paper highlights the complexity of balancing these parameters but does not provide robust, practical guidelines for optimal tuning.

Different data distributions and kernel eigenstructures result in non-uniform impacts of the tuning parameters, complicating the tuning process further. It is better the paper could consider more general date distributions in Assumption 2 or at least can provide some insights on how the results may vary given different date distributions.

**Questions:**

Can you explain the theoretical basis for the catastrophic overfitting observed with certain bandwidth scalings and the choice of kernels intuitively?

What are the practical guidelines for choosing the bandwidth parameter in real-world applications since there is no experiment?

How does the proposed method perform across different types of data distributions besides Assumption 2?

**Limitations:**

Yes

---

> ### Author Rebuttal · Authors · 2024-08-06
>
> We thank the reviewer for their thoughtful feedback. Below we address the main comments:
> 1. Regarding the experiments:  There is ample empirical evidence about the eigenframework holding well for the Gaussian kernel. For example, Figure 1 in [38] shows the predicted vs. true test risk for Gaussian KRR (data is from MNIST), and Figure 5 in [7] shows the true vs. predicted risk for Gaussian KRR with data uniform on the sphere (which is the setup we consider here). In both of these plots, the prediction matches closely with the true test risk. Furthermore, in the attached PDF we also provide empirical evidence that the claim for the case of fixed dimension holds. We plot the dependence of the test error on sample size for the Gaussian kernel ridgeless (interpolating) predictor. In Figure a, we consider the first case of Theorem 1, namely when the bandwidth $\tau_m = o(m^{-\frac{1}{d-1}})$. It shows that the test error tends to something equal to or larger than the test risk of the null predictor. In Figure b, we consider the second case, i.e $\tau_m = \omega(m^{-\frac{1}{d-1}})$ and the plot shows that the test error increases with sample size, which aligns with the claim that it diverges as the sample size $m$ increases. In Figure c, we consider the case when $\tau_m = \Theta(m^{-\frac{1}{d-1}})$ and the plot shows that for some noise level (which is high enough), the predictor is worse than the null predictor, as predicted by our theorem. We will add these simulations to the final version of the paper.
>
> 2. Intuitive explanation for the catastrophic behavior observed: As discussed in [24], the functions learned by the Gaussian kernel are too smooth, so they overfit the noise harmfully. On the other hand, the authors construct a “spiky-smooth” kernel that can exhibit benign overfitting in a fixed dimension.
>
> 3. Practical guidelines for optimal parameter tuning: Since the result for fixed dimension is negative, i.e. we show that the overfitting is catastrophic or tempered and that the predictor is almost always worse than the null predictor, it’s not expected that the Gaussian ridgeless regression would be successful in practice. The guideline would be to use an optimally tuned ridge and to avoid interpolation when using Gaussian KRR.
>
> 4. Different data distributions: As briefly mentioned in lines 142-145, the same result extends to more general distributions, namely uniform on a manifold that is diffeomorphic to the sphere. So the result holds for more general data distributions as well. Furthermore, considering inner product kernels with respect to the uniform distribution on the sphere is very common in many theoretical works, such as [4,37,38,61], just to mention a few. We agree with the reviewer that extending the results for additional distributions is an interesting future direction.

---

> > ### Comment · Reviewer_cvRM · 2024-08-13
> >
> > Thanks for the clarification and the attached PDF. I have increased my scores.

---

### Official Review · Reviewer_Ymgo · 2024-07-11

**Soundness:** 3
**Presentation:** 3
**Contribution:** 3
**Rating:** 7
**Confidence:** 3

**Summary:**

The authors derive estimates of the population risk of the kernel (ridgeless) regression estimator for spherical data when either the kernel bandwidth or the dimension of the data depends
on the number of training points $m$. To do so, authors rely on the eigenframework and prior work, which provides (under assumptions on the distributions of the random variable $\phi(X)$ where $\phi$ is the kernel feature map and $X$ is the input random variable) formulas for the population risk of the kernel ridge regression estimator as a function of the eigenvalues of the kernel integral operator. Using these formulas, authors then derive nonasymptotic estimates of these quantities in order to establish the limit value of the risk, as well as its dependency w.r.t $m$ and $d$. They obtain that

- for a fixed dimension, overfitting is always harmfull (tempered or catasrophic) unless the bandwidth scales in precisely in $m^{-1/d-1}$ in which case overfitting is shown to be tempered for large enough noise in the data (Theorem 1)
- for a fixed bandwidth, but varying dimensionthey derivelower and upper bound on the population risk. This allows them to
	- recover known results stating that when the dimension size increases polynomially with sample size, overfitting is benign or tempered
	- show that when the dependency is logarithmic, overfitting is not benign (under additional assumptions verified for the gaussian kernel)
	- show that when the dependency is subpolynomial $\exp(\sqrt{m)$ scaling, overfitting can be benign

**Strengths:**

The paper is clear, with a good effort spent on litterature review and comparison with prior work.

The main technical contributions seem to be bounds on the spectra of dot-product kernels on the sphere, which if novel (I am not an expert of this domain) could be of independent interest and are welcomed.

**Weaknesses:**

- This paper investigates the behavior of **ridgeless** (e.g. no RKHS norm penalty) kernel estimators, something which should be made clearer.  For example, kernel ridge regression is part of the paper title, whereas the paper does not study kernel ridge regression.
- The eigenframework which they use rely on a number of assumptions to be valid.

**Questions:**

In the analysis, the data is assumed to be on the sphere. Can the authors comment on how they expect the results to change for data
  distributed on the whole of $\mathbb  R^d$?


Typos:

l 294
l 298

**Limitations:**

Mentioned in weaknesses.

---

> ### Author Rebuttal · Authors · 2024-08-06
>
> We thank the reviewer for their positive feedback. Below we address the main comments:
>
> 1. Ridge vs Ridgeless: We will change the title to use “ridgeless” instead and reword other relevant parts of the paper.
>
> 2. Assumptions for the eigenframework: There is ample empirical evidence that the eigenframework holds well for the Gaussian kernel. For example, Figure 1 in [38] shows the predicted vs. true test risk for Gaussian KRR (using MNIST data) and Figure 5 in [7] shows the true vs. predicted risk for Gaussian KRR with data uniform on the sphere (which is the setup we consider here). In both of these plots, the prediction matches closely with the true test risk.
>
> 3. Answer to the reviewer's question: The Gaussian kernel will essentially zero out contributions of points that are far away, so as long as the bandwidth is small enough and the sample size is sufficiently large, the d-sphere will “look” the same as $\mathbb R^d$. So we expect that the result stays the same for other measures on $\mathbb R^d$, but we leave this question for further research.

---

> > ### Comment · Reviewer_Ymgo · 2024-08-09
> > **Thank you for your response**
> >
> > Thank you to the authors for their response.
> >
> > Regarding 2 - reading again that section of the paper, it think it would help guide the reader unfamiliar with this literature if the authors explained how the predicted risk expression is obtained. For instance, quoting [38], "[BCP20] presents two different approaches to obtain this analytical expression: a continuous approximation to the learning curves inspired by the Gaussian process literature [Sol01], and replica method with a saddle-point approximation". A sentence along these lines could be helpful to make this formula less "magic".

---

> > > ### Author Response · Authors · 2024-08-11
> > >
> > > We thank the reviewer for additional comments. We will update the update the paper to include this explanation.

---

### Author Rebuttal · Authors · 2024-08-07

We thank all of the reviewers for their thoughtful feedback.
In the attached PDF, we provide empirical evidence that our predictions are correct. We focus on the finite-dimensional case because of computational constraints. In Figure a, we consider the first case of Theorem 1, namely when the bandwidth $\tau_m = o(m^{-\frac{1}{d-1}})$. It shows that the test error tends to something equal to or larger than the test risk of the null predictor. In Figure b, we consider the second case, i.e $\tau_m = \omega(m^{-\frac{1}{d-1}})$ and the plot shows that the test error increases with sample size, which aligns with the claim that it diverges as the sample size $m$ increases. In Figure c, we consider the case when $\tau_m = \Theta(m^{-\frac{1}{d-1}})$ and the plot shows that for some noise level (which is high enough), the predictor is worse than the null predictor, as predicted by our theorem. We will add these simulations to the final version of the paper.

---

### Comment · Area_Chair_Kut5 · 2024-08-08
**Please read rebuttal and comment**

Dear reviewers,

could you have a look at the authors response and
comment on them if you have done so, yet. So far,
we have quite a bit of a spread in the opinions, so
we need a discussion to converge.

thanks in advance
your area chair

---

### Author Response · Authors · 2024-08-11
**Regarding the contribution of the paper**

We thank the reviewers for their continued feedback.
We feel some reviewers are dismissive of the contribution of analyzing the Gaussian kernel under the eigenframework. Indeed, we are relying on the eigenframework predictions, which are a powerful (but non-rigorous) tool, and proving the eigenframework predictions for the Gaussian kernel would be very important (and difficult) contribution. We acknowledge that understanding the limitations of eigenframework is an important research direction, but we point to both empirical [7,38] and theoretical evidence [38] that the eigenframework holds well for the Gaussian kernel.


Taking an extreme analogy, we feel that dismissing work because it relies on the eigenframework is analogous to dismissing hardness results relying on NP!=P because "the hard part was done by Cook, Lewin and Karp” or “more interesting is to prove NP!=P", or asking if the hardness result can be relaxed to "avoid this assumption". This analogy is extreme and we are not saying the eigenframework is at the same status as NP!=P, but more direct analogies are other papers relying on the eigenframework (e.g. [1, 7, 12, 25, 47, 54, 63]), or with a more historical perspective, the entire body of work relying on the Parisi predictions for the Sherrington-Kirkpatrik Ising Model, a spin glass model (e.g. by Mazard, Parisi, Virasoro and many others), which was only much later proven rigorously by Talagrand.


Yes, we are using a tool (the eigenframework predictions) that involved some heavy lifting, but that is the point of a tool - to allow for simplified further analysis.  The analysis we do using this tool is highly non-trivial at the technical level and is not just a simple application of Zhou et al 2023 [63]. First, for fixed dimension, our technical contribution is deriving the dependence of the eigenvalues of the Gaussian kernel on the bandwidth (Theorem 8) and hence understanding how the test risk depends on the bandwidth (Theorem 1). Second, for increasing dimension, our technical contribution is deriving upper and lower bounds for the test risk that work for any scaling of the dimension and sample size (Theorems 2 and 3), by introducing the “upper and lower index” (Definition 1).


We agree that proving the eigenframework predictions rigorously for a Gaussian kernel (i.e. lifting the GUA assumptions) would indeed be a strong contribution, but we are very explicit on exactly what we rely on, what is new, what is rigorous, and what is an assumption. We do not think the fact that we do not prove all of this in a single paper should diminish the contribution of what we actually do.

---

### Decision · Program_Chairs · 2024-09-25

**Decision:**

Accept (poster)

**Comment:**

The paper investigates how the "predicted risk" of kernel rigdeless
regression behaves for Gaussian kernels on the sphere if the width
changes with the sample size. The first main result shows that
catastrophic or tempered behavior of this risk surrogate occurs depending
on how the width changes. The second main result establishes a
similar result for more general kernels (without a width parameter)
if the dimension increases.

The paper enjoyed serious reviews as well as a productive rebuttal phase.
Most concerns centered around: a) the used eigenframework,
b) the Gaussian universality assumption, and c) the novelty of
the technical contribution. In view of a) the authors argued with a list
of recent papers working with the same assumption. However, about half
of these papers have not been published, yet, and hence it is probably
too early for a decisive conclusion about this framework. However, I
would argue for an "in dubio pro accept" approach as the alternative
(if applied consistently in all areas of ML) would substantially slow
down scientific progress. Issue b) is actually closely related to a)
for the paper under review as it provides a "justification" for a).
In view of c) the authors made a convincing case that they do not
simply apply the work of others.

Overall, I agree with the outcome of the discussion which went in favor
of accepting the paper despite the fact that some parts (e.g. Example 1)
could certainly be improved or clarified.